# TIGIT limits immune pathology during viral infections

Michelle Schorer[1], Nikolas Rakebrandt[1], Katharina Lambert [1], Annika Hunziker[2], Katharina Pallmer[3], Annette Oxenius [3], Anja Kipar [4], Silke Stertz[2] & Nicole Joller [1✉]

Co-inhibitory pathways have a fundamental function in regulating T cell responses and control the balance between promoting efficient effector functions and restricting immune pathology. The TIGIT pathway has been implicated in promoting T cell dysfunction in chronic viral infection. Importantly, TIGIT signaling is functionally linked to IL-10 expression, which has an effect on both virus control and maintenance of tissue homeostasis. However, whether TIGIT has a function in viral persistence or limiting tissue pathology is unclear. Here we report that TIGIT modulation effectively alters the phenotype and cytokine profile of T cells during influenza and chronic LCMV infection, but does not affect virus control in vivo. Instead, TIGIT has an important effect in limiting immune pathology in peripheral organs by inducing IL-10. Our data therefore identify a function of TIGIT in limiting immune pathology that is independent of viral clearance.

[1] Institute of Experimental Immunology, University of Zurich, Winterthurerstrasse 190, 8057 Zurich, Switzerland. [2] Institute of Medical Virology, University of Zurich, Winterthurerstrasse 190, 8057 Zurich, Switzerland. [3] Institute of Microbiology, ETH Zurich, Vladimir-Prelog-Weg 1-5/10 8093, Zurich, Switzerland. [4] Laboratory for Animal Model Pathology, Institute of Veterinary Pathology, Vetsuisse Faculty, University of Zurich, Winterthurerstrasse 268, 8057 Zurich, Switzerland. ✉email: nicole.joller@immunology.uzh.ch

Chronic viral infections represent a major burden on human health, and their effective treatment remains an unsolved challenge. One of the most striking features of such chronic diseases is that effector T cells enter a state of dysfunction also referred to as T cell exhaustion[1,2]. T cell exhaustion is characterized by the gradual loss of T cell effector functions as a consequence of continuous and prolonged antigen exposure[3,4]. Exhausted T cells display highly upregulated and sustained expression levels of co-inhibitory receptors, such as PD-1, Lag-3, and Tim-3, fail to proliferate and show a defective pro-inflammatory cytokine response[5]. Moreover, immune-modulatory cytokines like IL-10 also play an important role in disease progression during chronic infection, as it has been shown that polymorphisms in IL-10 or IL10RA are associated with increased susceptibility to chronic HCV and HBV infection[6,7]. Importantly, many of these characteristics are shared by dysfunctional T cells that arise during the development of cancer, and targeting co-inhibitory receptors has emerged as a potent immune-modulatory strategy in order to "revitalize" the anti-tumor response for the treatment of such conditions.

TIGIT (T cell immunoglobulin and ITIM domain) is a co-inhibitory receptor that acts as an important immune checkpoint, as it limits both T cell-driven inflammation and T cell and NK cell-dependent anti-tumor immunity[8,9]. TIGIT is expressed on activated T cells, regulatory T cells, and NK cells and binds two ligands CD155 (PVR) and CD112 (PVRL2 or Necl5), that are expressed on antigen-presenting cells (APCs)[10,11] and tumor cells[12,13]. CD226, the co-activating counterpart, competes with TIGIT for the same ligands[14], a feature which is reminiscent of the CTLA-4/CD28 pathway. TIGIT is known to exert its immune suppressive function through various modes of action, including direct and indirect inhibition of T cells. It was shown that T cell intrinsic TIGIT signaling directly inhibits T cell activation and proliferation by attenuating TCR signaling[9]. Moreover, it was demonstrated that TIGIT expression protects cells from NK-mediated killing[10], which highlights its role in promoting tumor progression. Regulatory T (Treg) cells constitutively express high levels of TIGIT at steady state[11,15], which equips them with superior suppressive capacity and indirectly suppresses effector T cell responses[16]. In addition, TIGIT binding to its cognate receptor CD155 on dendritic cells leads to the indirect suppression of T cell responses via induction of the immune suppressive cytokine IL-10 in DCs[11]. Importantly, loss of TIGIT results in hyperproliferative and inflammatory T cell responses and a marked reduction in IL-10[9,14]. IL-10 itself is a key regulator of immune responses during infections, where it plays a two-sided role by both promoting pathogen persistence[17–19] as well as limiting excessive Th1 and CD8+ T cell responses that cause immune pathology[20,21]. Incidentally, many chronic or latent human viruses have evolved to encode IL-10 homologs in order to evade immune control[22]. Nevertheless, the resolution of inflammatory processes relies on the tightly controlled balance between pro-inflammatory and regulatory immune responses, which is especially evident during the pathogenesis of neurotropic protozoa such as Plasmodium falciparum, Trypanosoma cruzi, or Toxoplasma gondii, where IL-10 signaling was demonstrated to restrict cytokine-mediated immune pathology[23,24]. TIGIT blockade or deficiency has been shown to lead to the breakdown of peripheral tolerance in HBsAg-tg mice, resulting in the development of hepatitis and fibrosis[25]. At the same time, TIGIT upregulation was shown to support tissue regeneration in vivo by negatively regulating NK cell activation[26]. Thus, a better understanding of immune-modulatory receptors and their impact on both pathogen control as well as maintaining tissue protection is crucial, especially in the view of their increased use as checkpoint inhibitors in the clinics.

As TIGIT ligation is functionally linked to IL-10 expression that is known to both promote virus persistence in vivo, but also limit adverse immunopathological damage, the TIGIT pathway might represent an important regulatory gatekeeper for the control of viral infections. Therefore, we investigated the role of TIGIT in modulating T-cell responses following chronic LCMV infection as well as its contribution to restricting immune pathology. Here, we report that TIGIT modulation effectively alters the cytokine profile and IL-10 expression levels in vivo. However, TIGIT blockade or stimulation alone is not sufficient to promote virus clearance or persistence but rather plays an important role in limiting immune pathology in peripheral organs in an IL-10-dependent manner.

## Results

**TIGIT is upregulated during chronic LCMV infection.** In a first line of investigation, we sought to identify whether TIGIT is expressed on exhausted T cells upon chronic viral infection with LCMV clone 13, the classical model to study T cell exhaustion and viral persistence[4,5]. We found that TIGIT is indeed highly expressed during the course of chronic LCMV clone 13 infection on both CD4+ and CD8+ T cells as well as on Tregs (Fig. 1a–c, h), the latter displaying the highest expression of TIGIT throughout the course of infection. TIGIT expression was particularly high on exhausted PD-1+ Tim-3+ CD8+ T cells (Fig. 1f), but not on PD-1− Tim-3− CD8+ T cells. We next asked whether TIGIT was also upregulated on antigen-specific T cells that are known to become exhausted during chronic viral infections. In order to follow antigen-specific T cells, Ly5.1+ TCR transgenic CD8+ and CD4+ T cells specific for the immuno-dominant gp33 (P14) or gp61 (Smarta) peptides were transferred into wild-type (WT) C57BL/6 mice prior to LCMV clone 13 infection. Indeed, antigen-specific T cells expressed high levels of TIGIT during the chronic phase of the infection starting from day 20 p.i. (Fig. 1d, e), suggesting that like PD-1[27,28], TIGIT expression is maintained in response to continuous TCR stimulation. Given that T cell-derived IL-10 was shown to contribute to T cell suppression and viral persistence in the LCMV model[18,29], we next tested whether TIGIT expression correlates with IL-10 production in vivo. We used Thy1.1-IL-10 reporter mice that carry a bacterial artificial chromosome Il-10 promotor transgene with a Thy1.1 cDNA insertion in order to reliably identify IL-10-producing cells[30]. In line with TIGIT being expressed on exhausted cells, TIGIT+ CD8+ T cells displayed significantly higher levels of IL-10 on day 30 p.i. than TIGIT− CD8+ T cells (Fig. 1g). Taken together, these findings suggest that TIGIT marks a PD-1+ IL-10-producing exhausted CD8+ T-cell population that arises during chronic LCMV infection.

**TIGIT modulates co-inhibitory receptors on CD8+ T cells.** In order to determine the functional contribution of the TIGIT pathway toward promoting T cell exhaustion during chronic LCMV infection, we targeted TIGIT in vivo by using the blocking anti-TIGIT antibody (clone 1B4) we have generated and characterized previously[31]. Chronically infected C57BL/6 mice were continuously treated with either anti-TIGIT or mouse IgG1 control antibodies starting on the day of infection. We observed that TIGIT blockade significantly altered the exhaustion phenotype of CD8+ T cells. During the chronic phase of the infection (day 30 p.i.), CD8+ T cells from anti-TIGIT-treated mice displayed markedly lower PD-1 and Tim-3 expression levels than controls (Fig. 2a, c). Decreased expression of PD-1, and Tim-3 on CD8+ T cells, was also detectable during early phases of LCMV clone 13 infection and remained considerably reduced until day 40 p.i. (Supplementary Fig. 1A). PD-1 expression was also

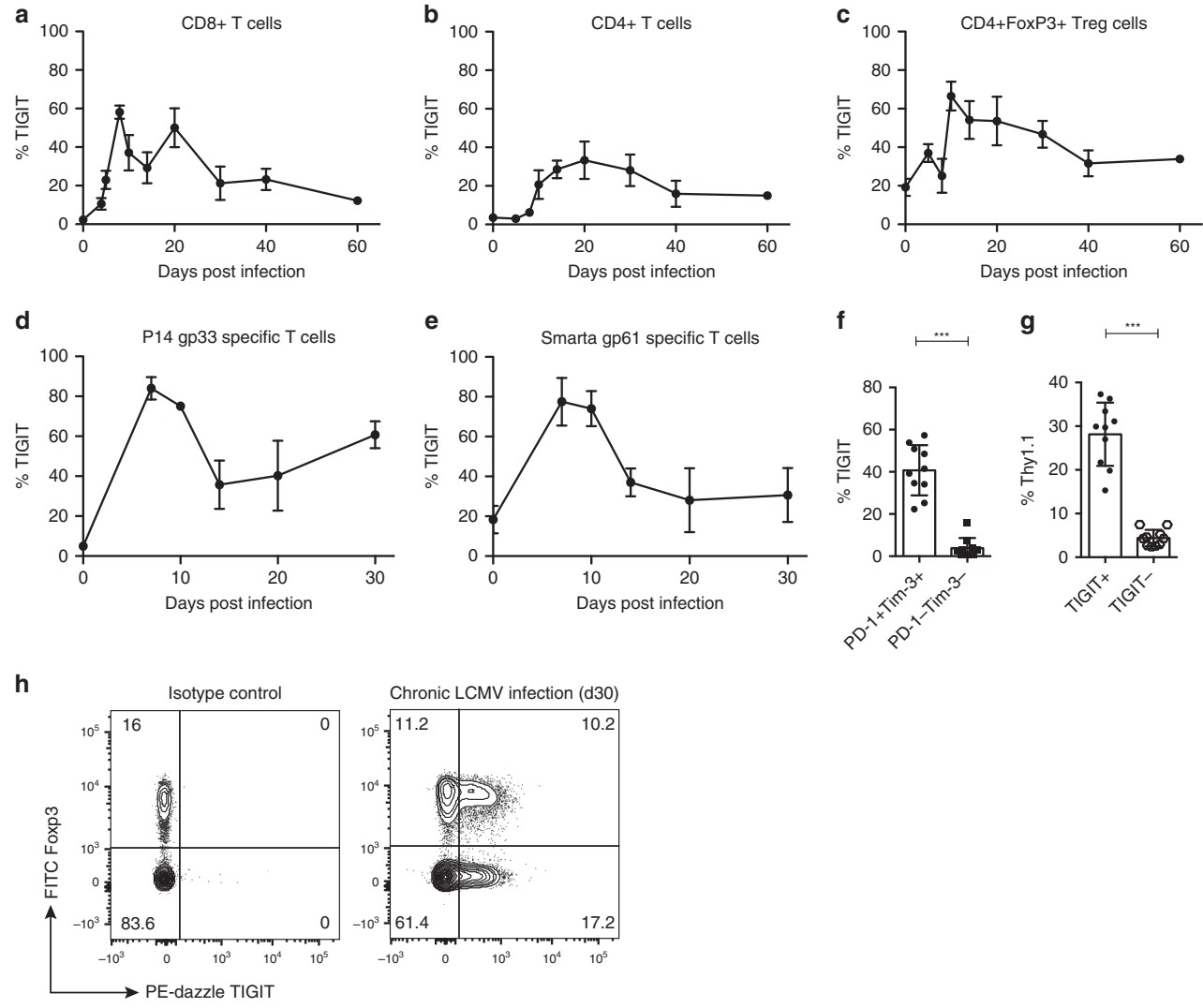

**Fig. 1 TIGIT expression during chronic LCMV clone 13 infection.** TIGIT expression on splenic T cells was analyzed after chronic LCMV clone 13 infection ($2 \times 10^6$ FFU) using flow cytometry. Expression over time on (**a**) CD8+ T cells, (n = 2–16), (**b**) CD4+ T cells (n = 2–17), and (**c**) CD4+FoxP3+ regulatory T cells (n = 2–13) is shown. TIGIT expression on (**d**) P14 Ly5.1+ TCR transgenic CD8+ T cells specific for gp33, (n = 4–6) and (**e**) Smarta CD4+ T cells specific for gp61 (n = 4–7). **f** TIGIT expression on exhausted (PD1+Tim-3+) or non-exhausted (PD1−Tim-3−) CD8+ T cells on day 30 p.i. (n = 10). **g** IL-10-Thy1.1 expression in Thy1.1-IL-10 reporter mice by TIGIT+ and TIGIT- CD8+ T cells on day 30 p.i. (n = 10). **h** Representative FACS plots of TIGIT expression on T cells on day 30 p.i. with chronic LCMV clone 13. Data represent 2–3 independent experiments. Each symbol in the scatter plot represents an individual mouse, and bar graphs indicate the mean value ± SD. Statistical values $p < 0.05$ (*), $p < 0.01$ (**), $p < 0.005$ (***) or ns (not significant, $p > 0.05$) determined by Welch's $t$ test.

significantly decreased on CD4+ T cells, however, only during early stages of infection (Supplementary Fig. 1B), while the PD-1 expression on regulatory T cells remained unchanged over the course of chronic infection. The in vivo anti-TIGIT antibody (clone 1B4) was already shown to be non-depleting after immunization with MOG peptide[31] and we confirmed these findings in chronic LCMV infection (Supplementary Fig. 1C). Because lack of TIGIT signaling could also have a negative impact on myeloid cells that express the ligand, we quantified the abundance of various populations of antigen-presenting cells in the spleen (Supplementary Fig. 1D, E), but could not detect any noticeable differences, neither regarding their frequency nor their absolute numbers. Moreover, we analyzed the NK cell phenotype over the course of acute and chronic LCMV infection with and without anti-TIGIT Ab administration and found them to be comparable (Supplementary Fig. 2A–E).

In order to determine whether TIGIT might be able to actively promote T-cell exhaustion, we infected WT mice with an intermediate dose of LCMV clone 13 ($1 \times 10^5$ FFU), which results in an acute infection that is cleared within 10 days and treated them with either an agonistic anti-TIGIT antibody (1G9) or IgG1 isotype control. Indeed, antibody-mediated TIGIT engagement resulted in increased PD-1 and Tim-3 expression on CD8+ T cells on day 14 p.i. (Fig. 2b, d). These results demonstrate that TIGIT modulation has an impact on the exhaustion phenotype of T cells.

**TIGIT correlates with IL-10 production in vivo**. Given that IL-10 was shown to contribute to viral persistence in vivo[19,29,32] and that TIGIT signaling induces the production of IL-10 both directly and indirectly[11,31], we speculated that TIGIT might hold a central role in contributing to viral persistence through its ability to induce IL-10. To further investigate this link between TIGIT and IL-10 expression in vivo, we chronically infected Thy1.1-IL-10 reporter with LCMV clone 13 and again treated

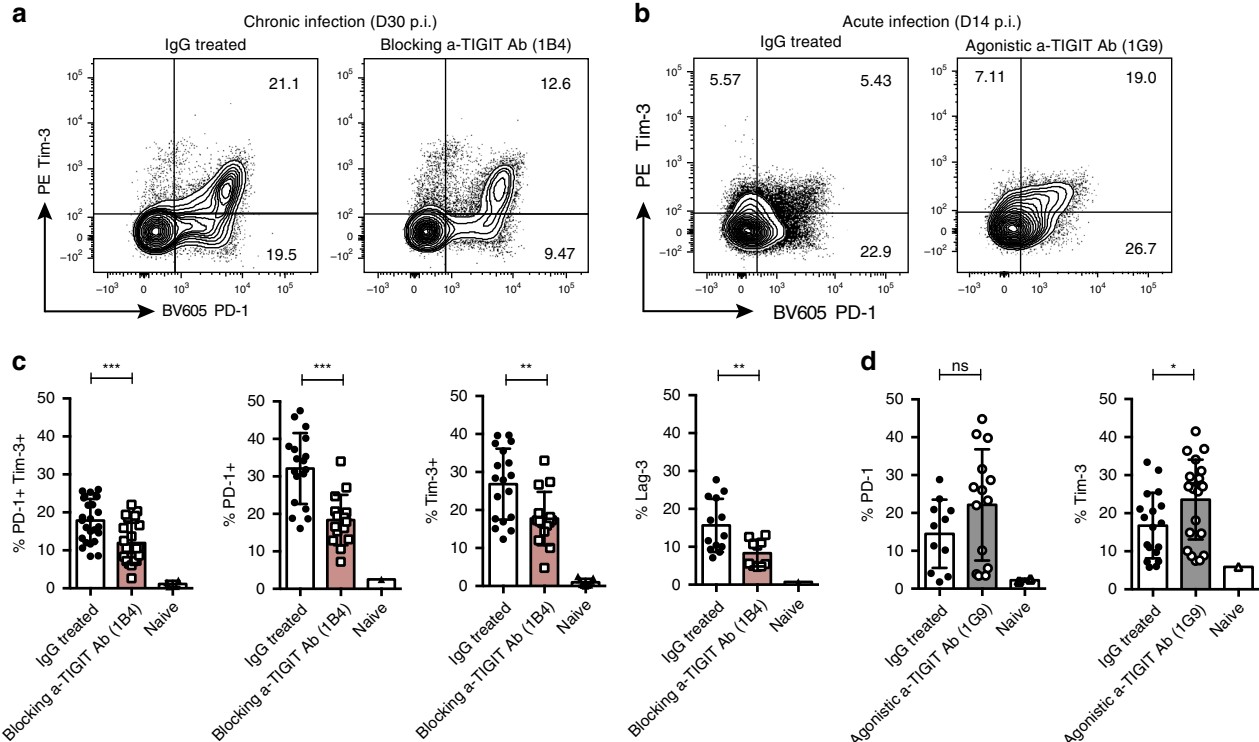

**Fig. 2 In vivo TIGIT modulation alters co-inhibitory receptor expression on T cells after LCMV infection.** C57BL/6 mice were infected with either $2 \times 10^6$ FFU LCMV clone 13 i.v. (red, chronic) or $1 \times 10^5$ FFU LCMV clone 13 i.v. (gray, acute) and treated with 100 µg of blocking anti-TIGIT Ab (1B4, chronic infection), agonistic anti-TIGIT Ab (1G9, acute infection), or mouse IgG1 i.p. Representative FACS plots (**a**, **b**) and summary data (**c**, **d**) of co-inhibitory receptor expression on splenic CD8$^+$ T cells after (**a**, **c**) chronic LCMV infection (day 30, n = 10–25), and (**b**, **d**) acute LCMV infection (day 14, $n$ = 11–22) are shown. Data represent pooled data from four to five independent experiments. Each symbol in the scatter plot represents an individual mouse, and bar graphs indicate the mean value ± SD. Statistical values $p < 0.05$ (*), $p < 0.01$ (**), $p < 0.005$ (***), or ns (not significant, $p > 0.05$) determined by Welch's $t$ test.

them with blocking anti-TIGIT antibody (1B4) or isotype control. TIGIT blockade resulted in a decrease in the frequency of IL-10-Thy1.1$^+$ CD8$^+$ T cells (Fig. 3a). Vice versa, TIGIT engagement in the course of acute LCMV infection using the agonistic anti-TIGIT antibody led to significantly increased frequencies of both IL-10-Thy1.1$^+$ CD8$^+$ T cells and IL-10-Thy1.1$^+$ CD4$^+$ T cells (Fig. 3a). Yet, TIGIT modulation did not affect overall numbers of IL-10-Thy1.1$^+$ T cells in the spleen in any of the infection models (Fig. 3b). Nevertheless, evaluation of serum IL-10 levels in chronically infected mice revealed an early drop of IL-10 around day 14 p.i. in mice treated with blocking anti-TIGIT antibody (1B4) (Fig. 3c). To confirm these observations, we analyzed supernatants of anti-CD3 in vitro re-stimulated splenocytes from chronically infected mice (day 30 p.i.). In line with our previous findings, TIGIT blockade correlated with significantly decreased levels of IL-10 (Fig. 3d). In turn, engagement of the TIGIT pathway in acutely infected mice resulted in elevated serum levels of IL-10 (Fig. 3e). Within the T cell compartment, only effector T cells displayed an altered IL-10 phenotype. In addition, we analyzed the Thy1.1 expression in myeloid cell populations during both acute and chronic LCMV infection. During early acute LCMV infection, we could not detect any significant differences regarding the frequency of IL-10-Thy1.1$^+$ cells between control and agonistic anti-TIGIT (1G9) antibody-treated mice (Supplementary Fig. 3A). On day 14 p.i., CD11b$^+$Ly6C$^+$ cells showed significantly higher expression of IL-10-Thy1.1 after treatment with the agonistic anti-TIGIT (1G9) Ab (Supplementary Fig. 3B), which mimics the findings observed in the T cell compartment. However, the overall frequency of IL-10-Thy1.1$^+$ cells was much smaller than in the T cell compartment. Similarly,

we did not observe any differences in IL-10-Thy1.1 expression during the early phase of chronic LCMV infection on myeloid cells (Supplementary Fig. 3C), but found a trend toward decreased expression of IL-10-Thy1.1 in CD11b$^+$Ly6G$^+$ cells during the chronic stage of the infection (Supplementary Fig. 3D), however, this did not reach significance. Collectively, these data reveal a functional link between TIGIT engagement and IL-10 induction during LCMV infection.

**TIGIT dampens pro-inflammatory cytokines in acute infection.** A cardinal feature of exhausted T cells is the hierarchical and progressive loss of pro-inflammatory cytokine production, where IL-2 secretion is abolished prior to TNF secretion, and T cell exhaustion finally becomes terminal when the ability to produce IFN-γ is lost[2,33]. Conversely, we and others have previously shown that agonistic anti-TIGIT antibodies reduce pro-inflammatory cytokine production, both in vitro and in vivo[9,14,31]. We thus next analyzed whether TIGIT can modulate the expression of pro-inflammatory cytokines in T cells during both acute and chronic LCMV infection.

Agonistic anti-TIGIT antibody (1G9) treatment in acutely LCMV infected mice resulted in markedly reduced production of IFN-γ, but not TNF-α in CD8$^+$ T cells (Fig. 4a–c). Similarly, CD4$^+$ T cells also produced significantly lower amounts of IFN-γ after anti-CD3 re-stimulation (data not shown). Furthermore, agonistic anti-TIGIT antibody (1G9) administration resulted in the pronounced inhibition of T cell function on day 5 p.i. as indicated by the decreased expression of activation markers and pro-inflammatory cytokine production measured by FACS, confirming its immune suppressive properties (Supplementary

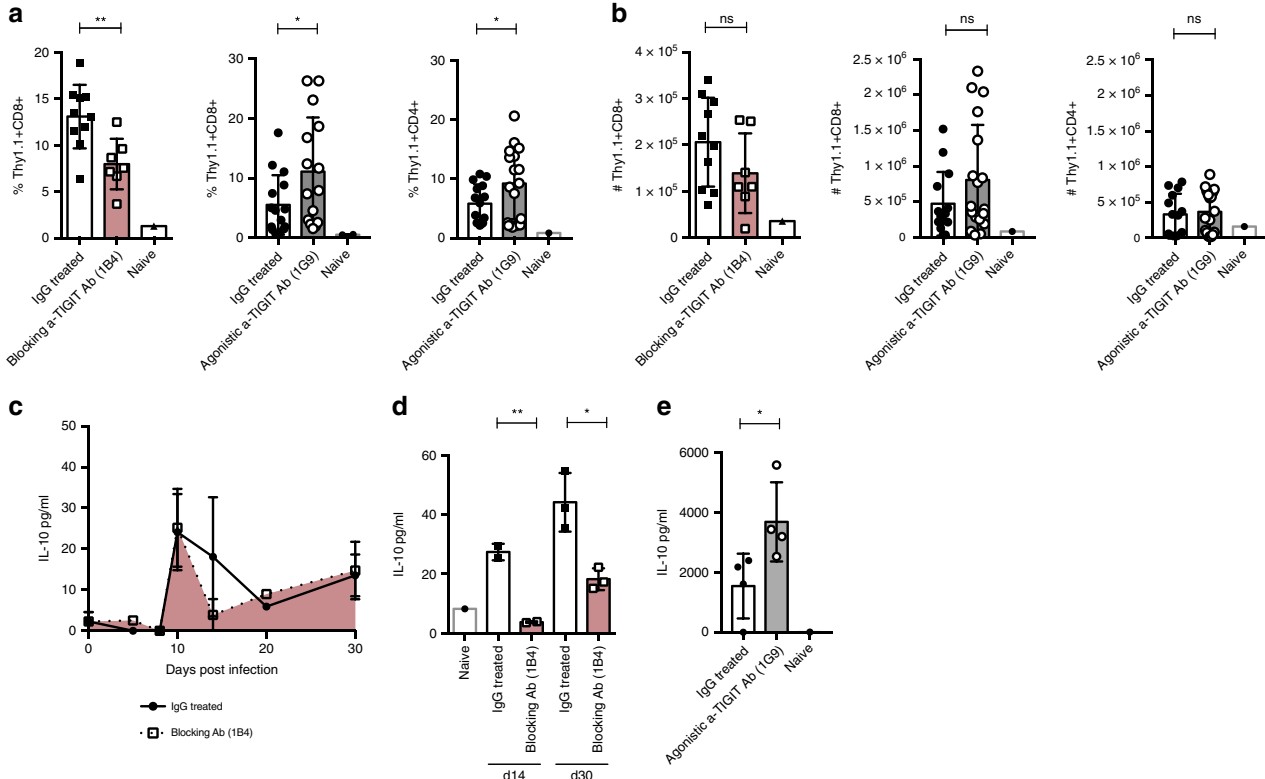

**Fig. 3 TIGIT modulation correlates with IL-10 production in vivo.** Thy1.1-IL-10 mice were infected with either $2 \times 10^6$ FFU LCMV clone 13 i.v. (red) or $1 \times 10^5$ FFU LCMV clone 13 i.v. (gray). Chronically infected mice (red) were treated with 100 μg of blocking anti-TIGIT (1B4) Ab or mouse IgG1 i.p., acutely infected mice (gray) were treated with 100 μg of agonistic anti-TIGIT (1G9) Ab or mouse IgG1 i.p. **a** IL-10-Thy1.1 expression on splenic CD8$^+$ on day 30 p.i. in chronically infected mice (red) and on CD8$^+$ and CD4$^+$ T cells on day 14 p.i. in acutely infected mice (gray) is shown ($n = 7$–17). **b** Absolute numbers of IL-10-Th1.1$^+$ CD8$^+$ on day 30 p.i. in chronically infected mice (red) and of IL-10-Th1.1$^+$ CD8$^+$ and IL-10-Th1.1$^+$ CD4$^+$ on day 14 p.i. in acutely infected mice (gray) are depicted ($n = 7$–13). IL-10 protein levels were measured by cytometric bead assay in (**c**) the serum of chronically (d30 p.i.) infected mice ($n = 1$–6 sera pooled from three mice) and (**d**) supernatants of ex vivo cultured splenocytes isolated on day 30 p.i. after chronic LCMV clone 13 infection re-stimulated with anti-CD3 ($n = 3$, pooled from nine mice) and (**e**) supernatants of ex vivo cultured splenocytes isolated on day 14 after acute LCMV clone 13 infection re-stimulated with anti-CD3 ($n = 4$). Pooled data from two to three independent experiments are shown. Each symbol in the scatter plot represents an individual mouse or pooled data point and bar graphs indicate the mean value ± SD. Statistical values $p < 0.05$ (*), $p < 0.01$ (**), $p < 0.005$ (***), or ns (not significant, $p > 0.05$) were determined by Welch's $t$ test.

Fig. 4A–D). In line with these results, TIGIT blockade yielded a modest but significant increase of TNF-α expression by CD8$^+$ T cells after re-stimulation with anti-CD3 (Fig. 4d, e). However, there was no difference in IFN-γ levels or in IL-2 production, which was below the detection limit (data not shown). Unexpectedly, CD8$^+$ T cells from control or anti-TIGIT antibody (1B4) treated chronically infected mice displayed no differences regarding TNF-α and IFN-γ production when re-stimulated in an antigen-specific manner (Fig. 4f). Taken together, these results demonstrate that TIGIT blockade during chronic LCMV infection only exerts a modest effect on pro-inflammatory cytokine production by T cells, whereas TIGIT stimulation during acute infection effectively inhibits the production of IFN-γ by both effector T cell subsets.

**Loss of virus-specific T cells upon TIGIT blockade.** Since virus-specific CD8$^+$ T cells play a critical role in virus clearance and were demonstrated to be most severely affected by exhaustion and deletion following high dose LCMV infection[34], we undertook further experiments to investigate the LCMV-specific T cell response. To this end, we adoptively transferred Ly5.1$^+$ TCR transgenic CD8$^+$ P14 cells into Ly5.2 recipient mice one day prior to chronic LCMV clone 13 infection. Strikingly, TIGIT blockade severely reduced the number of virus-specific P14 cells in the

blood of chronically infected mice (Fig. 5a). The same pattern was observed when endogenous CD8$^+$ T cells were stained with the gp33 peptide/MHCI tetramer against gp33 (Fig. 5b). Concurrently, the number of IFN-γ-producing P14 cells was significantly reduced early during chronic infection (Fig. 5c), but evened out by day 20 p.i. Furthermore, in contrast to total CD8$^+$ T cells the transferred P14 cells displayed elevated co-inhibitory receptor expression after TIGIT blockade (Fig. 5d), as well as slightly reduced pro-inflammatory cytokine production on day 30 after chronic LCMV challenge (Fig. 5e, f), indicative of a more exhausted phenotype. A population of TCF-1$^+$ CD8$^+$ T cells was previously shown to be critical to sustain the anti-viral immune response during chronic LCMV infection, as this subset serves as the source of cells that gives rise to differentiated effector like cells that respond to checkpoint blockade in vivo[35]. Interestingly, we found that TIGIT blockade leads to the loss of these TCF-1$^+$ CD8$^+$ T cells in both transferred antigen-specific and endogenous CD8$^+$ T cells (Supplementary Fig. 5A, B).

Given that the virus-specific cells seemed to be impaired following TIGIT blockade, but total T cells showed enhanced TNF-α production, we next addressed how inhibition of the TIGIT pathway would affect viral loads and persistence. We observed that TIGIT blockade failed to promote virus clearance in the spleen and kidneys of chronically infected animals (Fig. 5g, h). Similarly, we

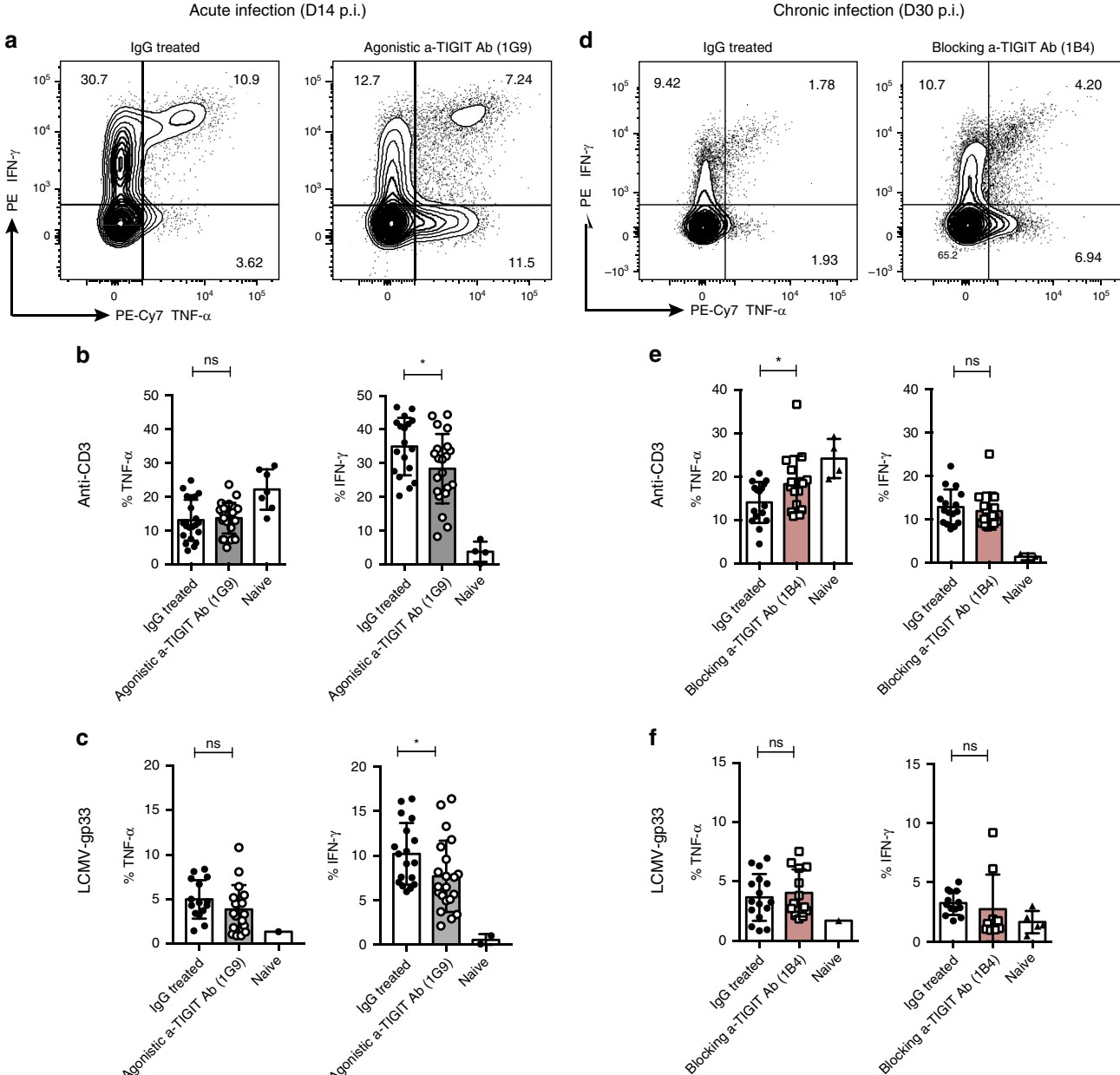

**Fig. 4 TIGIT ligation results in decreased pro-inflammatory cytokine production after acute infection.** C57BL/6 mice were infected with either $2 \times 10^6$ FFU LCMV clone 13 i.v. (red) or $1 \times 10^5$ FFU LCMV clone 13 i.v. (gray). Chronically infected mice (red) were treated with 100 μg of blocking anti-TIGIT (1B4) Ab or mouse IgG1 i.p., acutely infected mice (gray) were treated with 100 μg of agonistic anti-TIGIT (1G9) Ab or mouse IgG1 i.p. **a**, **d** Representative FACS plots displaying pro-inflammatory cytokine expression of splenic CD8$^+$ T cells after acute infection (gray) and chronic infection (red) (**b**) on day 14 p.i. with acute LCMV clone 13. FACS-based quantification of pro-inflammatory cytokine production by splenic CD8$^+$ T cells after anti-CD3 re-stimulation ($n = 18$–22) (**c**) on day 14 p.i. after gp33 re-stimulation ($n = 13$–20). FACS-based quantification of pro-inflammatory cytokine production by splenic CD8$^+$ T cells after (**e**) after anti-CD3 re-stimulation ($n = 15$–25) (**f**) after gp33 re-stimulation ($n = 9$–16). Each symbol in the scatter plot represents an individual mouse, and bar graphs indicate the mean value ± SD. Statistical values $p < 0.05$ (*), $p < 0.01$ (**), $p < 0.005$ (***), or ns (not significant, $p > 0.05$) determined by Student's $t$ test.

found that TIGIT stimulation during acute viral infection only temporarily impacts viral titers, as the virus was generally cleared by day 14 p.i. despite elevated virus loads in the spleen, but not the liver at an early time point (day 5 p.i.; Supplementary Fig. 4E–G). Collectively, these data suggest that TIGIT modulation alone is insufficient to affect viral persistence in vivo.

**TIGIT limits tissue damage in an IL-10-dependent manner.** Despite the fact that TIGIT modulation did not affect viral loads following acute LCMV infection by day 14 p.i., we noticed

marked differences regarding body weight loss between IgG1 and agonistic anti-TIGIT antibody (1G9) treated mice (Fig. 6a). Control mice showed a slight drop of around 8% of their initial body weight early during acute LCMV infection, which was not observed in agonistic anti-TIGIT antibody (1G9)-treated mice, the latter of which maintained or even gained weight over the course of infection. Because TIGIT engagement allows for the effective modulation of IL-10, which in turn plays an important role in the resolution of inflammation in various infectious as well as autoimmune models[36–38], we hypothesized that TIGIT-

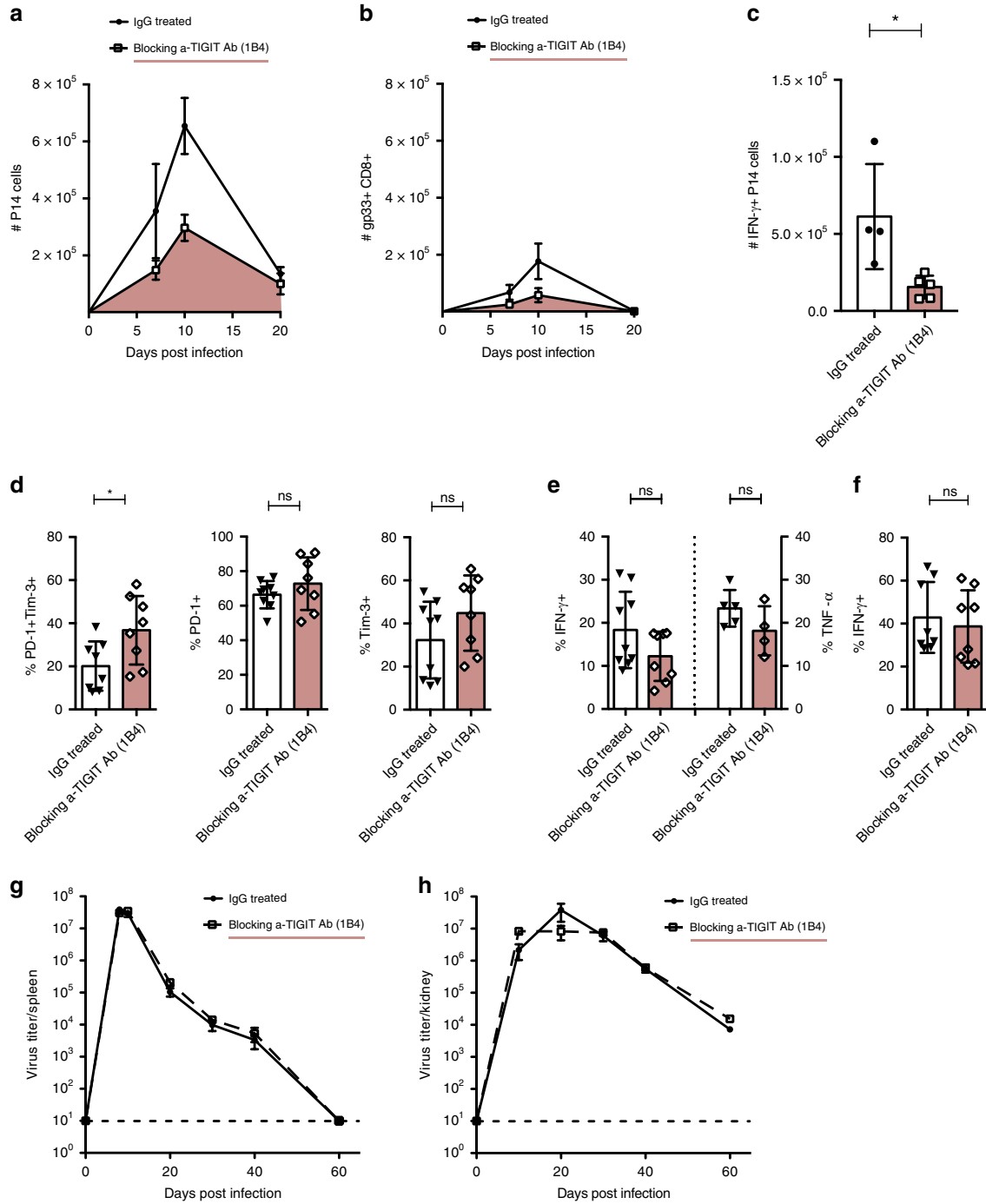

**Fig. 5 Loss of virus-specific T cells after TIGIT blockade during chronic LCMV infection.** 10'000 Ly5.1$^+$ P14 cells were transferred i.v. one day prior to chronic LCMV clone 13 infection and treatment with 100 μg of anti-TIGIT (1B4) Ab or mouse IgG1 i.p. FACS quantification of (**a**) absolute numbers of P14 Ly5.1$^+$ TCR transgenic CD8$^+$ T cells ($n = 4$–5) and (**b**) antigen-specific CD8$^+$ T cells by peptide/MHCI tetramer staining against gp33 ($n = 4$–5) in the blood. **c** Absolute numbers of IFN-γ producing P14 cells on day 10 were measured by FACS, (spleen, $n = 4$–5). **d** PD-1 and Tim-3 expression on P14 cells was quantified by FACS (spleen, day 20 p.i.; $n = 8$–9). **e, f** Pro-inflammatory cytokine production by TCR transgenic CD8$^+$ T cells quantified by FACS after anti-CD3 (**e**) or gp33 (**f**) re-stimulation was determined by FACS (spleen, day 20 p.i.; $n = 3$–9). Quantification of replicating virus particles by plaque assay in the spleen (**g**) and kidney (**h**) of chronically infected mice with and without anti-TIGIT treatment ($n = 10$). Each symbol in the scatter plot represents an individual mouse, and bar graphs indicate the mean value ± SD. Statistical values $p < 0.05$ (*), $p < 0.01$ (**), $p < 0.005$ (***),ns (not significant, $p > 0.05$) determined by Student's $t$ test.

mediated induction of IL-10 could protect against immune pathology. LCMV infection is known to cause T cell-dependent liver damage[39]. We thus sought to test if TIGIT stimulation might limit tissue damage by inhibiting T cell activation and infiltration in the liver. In order to characterize and quantify the immune pathological changes, we measured serum levels of aspartate aminotransferase (AST) and alanine transaminase (ALT), which are indicative of injured or damaged hepatocytes[40]. In line with our hypothesis, we found that TIGIT engagement resulted in significantly decreased AST and ALT levels on day 14 p.i. (Fig. 6b,

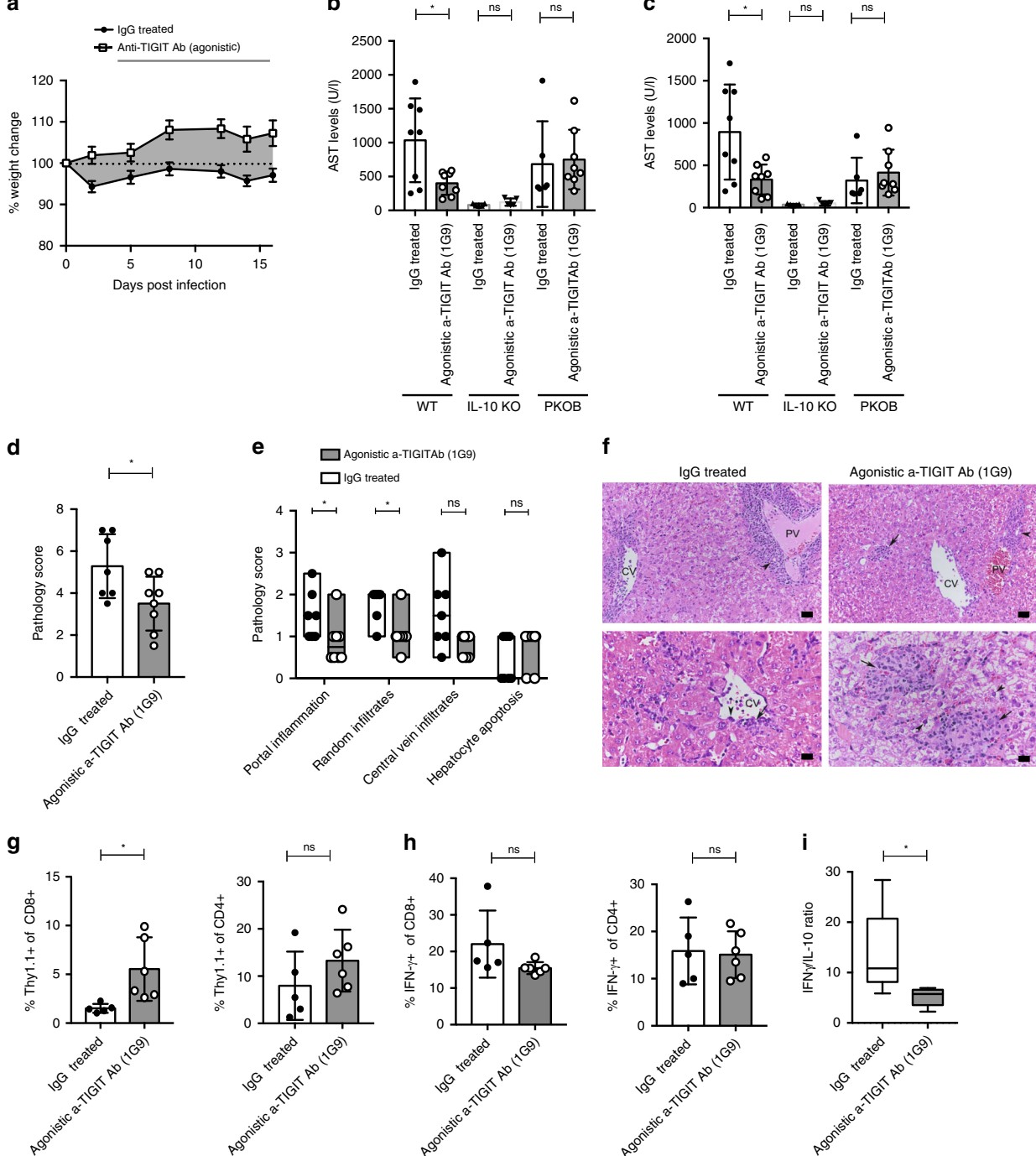

**Fig. 6 TIGIT engagement limits immune-mediated tissue pathology in the liver after acute virus challenge.** C57BL/6, Thy1.1-IL-10 reporter, IL-10 KO, or PKOB mice were infected with $1 \times 10^5$ FFU LCMV clone 13 i.v. and treated with 100µg control IgG1 or anti-TIGIT Ab (1G9) on days 0, 2, 4, and day 10 i.p. **a** The body weight of acutely infected mice over the course of infection with or without anti-TIGIT Ab (1G9) treatment is shown ($n = 10$). Serum levels of AST (**b**) and ALT (**c**) quantified by enzymatic activity assay on day 14 p.i. from WT, IL-10 KO, and PKOB mice are shown ($n = 5$–8). **d**–**f** Tissue damage was assessed in liver sections 14 days after infection using hematoxylin and eosin (H&E) staining. Overall pathology (**d**), classification of histopathological damages (**e**) and representative light microscopy images (**f**) are depicted ($n = 7$-8). Arrows indicate leukocyte recruitment and infiltration or dying hepatocytes. CV central vein, PV portal vein. Scale bar = 20 µm. **g**–**i** Cytokine production by T cells in the liver of IgG1 and anti-TIGIT Ab (1G9)-treated mice was determined by FACS on day 14 p.i. **g** Absolute numbers of IFN-γ+ CD8+ and CD4+ T cells after anti-CD3 re-stimulation. **h** Frequency of IL-10-Thy1.1+ CD8+ and CD4+ T cells (**i**) and the ratio of total IFN-γ+ to IL-10+ CD8+ T cells are displayed ($n = 5$-9). Pooled data from two independent experiments are shown. Each symbol in the scatter plot represents an individual mouse, and bar graphs indicate the mean value ± SD. Statistical values $p < 0.05$ (*), $p < 0.01$ (**), $p < 0.005$ (***), ns (not significant, $p > 0.05$) determined by Student's $t$ test (two experimental groups) or one-way ANOVA (more than two experimental groups). Histopathological data were evaluated using two-tailed Mann–Whitney test.

c). In order to better understand the effect of agonistic anti-TIGIT antibody (1G9) treatment, we undertook a histological examination of the livers of animals sacrificed on day 14 p.i. H&E staining of the livers revealed a similar pattern of disease in both experimental groups. We could observe on-going parenchymal damage, illustrated by the presence of scattered dying hepatocytes and evidence of active leukocyte recruitment and infiltration. The latter was indicated by the presence of leukocytes in the lumen of central veins and their rolling along endothelial cells. Leukocytes often appeared activated, with leukocyte infiltration stretching from central veins to hepatic cords (Fig. 6f). Leukocyte infiltration was also seen in portal areas and as random aggregates in the parenchyma. However, the magnitude of histopathological changes and overall disease score was significantly decreased in mice receiving agonistic anti-TIGIT antibody (1G9) treatment, especially regarding portal vein inflammation and central vein infiltration (Fig. 6d–f). Collectively, these findings link TIGIT to restricting immune-mediated liver damage during acute LCMV infection.

We next tested whether IL-10 is involved in mediating the reduced inflammatory response in agonistic anti-TIGIT treated mice and repeated the acute LCMV infection in IL-10 KO mice. In contrast to WT mice, agonistic anti-TIGIT treatment had no effect on AST or ALT levels in IL-10 KO mice (Fig. 6b, c). Serum AST and ALT levels in the IL-10 KO mice at this time point were generally very low compared with WT animals due to accelerated viral clearance and hence faster resolution of inflammation in IL-10 KO mice (Supplementary Fig. 6A–C, F, G). However, analysis of serum samples obtained on day 5 p.i. showed higher AST levels in IL-10 KO in comparison with WT mice (Supplementary Fig. 6D, E), which supports the notion that enzyme leakage from the liver occurs as a consequence of IL-10 deficiency, albeit following accelerated kinetics. In line with the results obtained on day 14, the liver protective effect of agonistic anti-TIGIT treatment was dependent on IL-10 as IL-10 KO mice showed no difference in AST/ALT levels upon treatment on day 5 p.i. (Supplementary Fig. 6D, E). An important mediator of tissue damage is perforin, which was shown to contribute to CD8+ T cell-dependent hepatitis in a model of virally induced immune pathology[41] and to CD8+ T cell-driven immunopathology in PD-L1-deficient mice upon LCMV infection[42]. Therefore, we speculated that TIGIT might be involved in regulating perforin-dependent tissue damage following acute virus infection. In order to test this, we infected PKOB mice with acute LCMV and either treated with control IgG1 or agonistic anti-TIGIT antibody (1G9). We found that the effect of TIGIT agonism is lost in PKOB mice, which suggests that the pathological process inhibited by TIGIT is perforin dependent (Fig. 6b, c).

To determine whether TIGIT stimulation results in local IL-10 production and an IL-10 dominated local immune environment, we next analyzed the immune infiltrates in the liver after acute LCMV challenge. Indeed, agonistic anti-TIGIT antibody (1G9)-treated mice showed a markedly increased frequency of IL-10-producing T cells in the liver (Fig. 6g). However, there was no difference regarding the overall number of IL-10-producing T cells (data not shown). Furthermore, no difference regarding the frequency of IFN-γ+ CD8+ T cells in the liver could be observed, irrespective of antigen-unspecific or -specific re-stimulation (Fig. 6h). More importantly, the ratio of IL-10+ to IFN-γ+ cells was strongly shifted toward an anti-inflammatory cytokine profile in the tissue in anti-TIGIT-treated groups (Fig. 6i), supporting the notion that TIGIT favors the generation of a regulated immune environment and limits immune-mediated tissue damage in an IL-10-dependent manner. What is more, we tested if blocking anti-TIGIT antibody (1B4) administration has the potential to exacerbate disease severity.

TIGIT blockade resulted in delayed recovery from chronic LCMV infection-induced body weight loss (Supplementary Fig. 7A). However, this did not go along with increased enzyme leakage from the liver (Supplementary Fig. 7B). Along the same lines, TIGIT blockade did neither lead to increased tissue pathology in any of the examined disease models, nor did it have an influence on virus loads (Supplementary Fig. 7C–F).

To further explore whether the tissue-protective effect of TIGIT was restricted to LCMV infection or whether it represents a general feature of the TIGIT pathway, we tested the effect of the agonistic anti-TIGIT antibody after influenza virus infection, a highly cytopathic virus. Anti-TIGIT and control-treated WT and IL-10 KO mice were infected with the mouse adapted H1N1 influenza A virus PR8, and the extent of tissue damage in the lungs was determined on day 8 p.i. In line with our findings in LCMV infection, TIGIT pathway engagement resulted in significantly decreased vascular leakage as indicated by the reduced levels of albumin-bound Evan's blue (EB) dye recovered from the lungs of anti-TIGIT-treated mice (Fig. 7a). Importantly, this effect was IL-10 mediated, as PR8 challenged IL-10 KO mice showed a reversed phenotype after agonistic anti-TIGIT antibody (1G9) treatment when compared with isotype controls (Fig. 7b). Moreover, agonistic anti-TIGIT antibody treatment led to a significantly decreased frequency of IFN-γ producing CD8+ T cells in the lungs of infected animals (Fig. 7c). At the same time, CD8+ and CD4+ T cells expressed markedly higher levels of IL-10 compared with IgG1-treated controls (Fig. 7d). Furthermore, total numbers of IL-10-producing CD8+ T cells were significantly increased in the lungs after TIGIT stimulation (Fig. 7e). Overall, the cytokine production in the lungs of anti-TIGIT antibody-treated mice was skewed toward an IL-10-dominated immune response, as the ratio of IL-10 to IFN-γ and TNF-α-producing T cells was greater in these animals (Fig. 7f). In addition, IL-10 protein levels were significantly elevated in lung homogenates of mice treated with agonistic anti-TIGIT antibody (1G9) compared with IgG1 controls (Fig. 7g). Importantly, anti-TIGIT treatment had no effect on influenza titers in both bronchoalveolar lavage (BAL) and lungs of infected animals throughout the course of PR8 infection (Supplementary Fig. 8A, B). PR8 infection lead to moderate vasculitis, which was most pronounced in IgG1-treated control mice. We also observed stronger evidence of on-going leukocyte recruitment into the tissue, which was most evident by the higher frequency and intensity of vasculitis in the control group (Supplementary Fig. 8C–E). Taken together, these findings further support the notion that the differences in tissue damage during PR8 infection were not a result of differential virus replication in the tissue, but rather the consequence of TIGIT-dependent immune modulation. In summary, these data reveal an important role of the TIGIT pathway in limiting immune-mediated tissue damage during acute virus infection in an IL-10-dependent manner.

## Discussion

Co-inhibitory pathways serve as central, negative regulators of immune cell functions, and have proven to be powerful targets for therapeutic immune modulation in diverse disease indications. The TIGIT inhibitory pathway is known to play an important role in limiting autoimmunity as well as negatively regulating anti-tumor responses. In this study, we investigated the significance of the TIGIT pathway in modulating T cell function during viral infections. We determined the impact of TIGIT blockade, as well as TIGIT engagement, on T cell function and phenotype during chronic and acute LCMV infections, respectively, and found that rather than promoting T cell dysfunction and viral persistence like other co-inhibitory receptors, TIGIT

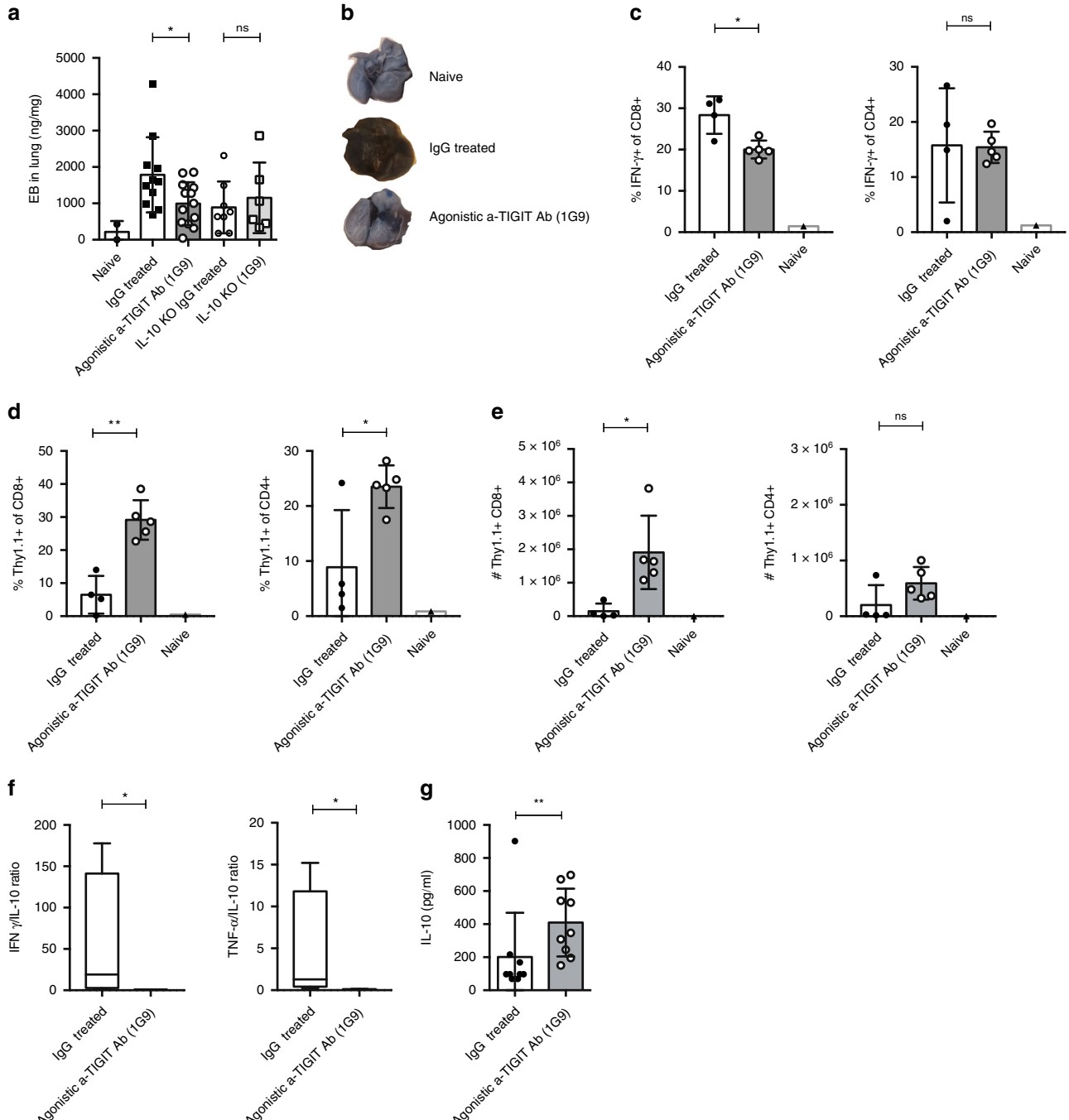

**Fig. 7 TIGIT stimulation protects from lung vascular leakage after influenza infection.** C57BL/6 mice and IL-10 KO mice were infected with 150 PFU PR8 virus i.n. and treated with 100 μg of control IgG1 or anti-TIGIT Ab (1G9) on days 0, 2, and 4 i.p. Evans blue (EB) dye was injected i.v. on day 8 p.i., followed by lung perfusion with PBS. **a** Colorimetric quantification of EB extravasation from the lung. The mean EB concentration in the lungs was normalized to the weight of the organ ($n = 6$–16). **b** Representative pictures of PBS perfused lungs of naive WT mice or mice with and without anti-TIGIT (1G9) Ab treatment on day 8 p.i. are shown. **c–f** The T cell phenotype was assessed on day 8 p.i. in the lung of IgG1 and anti-TIGIT (1G9) Ab-treated mice by FACS. **c** Frequency of IFN-γ$^+$ CD8$^+$ and CD4$^+$ T cells. **d** Frequency of IL-10-Thy1.1$^+$ CD8$^+$ and CD4$^+$ T cells. **e** Absolute numbers of IL-10-Thy1.1$^+$ CD8$^+$ and CD4$^+$ T cells and (**f**) the ratio of total IFN-γ$^+$ to IL-10$^+$ CD8$^+$ T cells and TNF-α$^+$ to IL-10$^+$ CD8$^+$ T cells are displayed ($n = 4$–5). **g** IL-10 protein levels were measured in whole-lung homogenates by cytometric bead assay ($n = 9$). Each symbol in the scatter plot represents an individual mouse and bar graphs indicate the mean value ± SD. Statistical values $p < 0.05$ (*), $p < 0.01$ (**), $p < 0.005$ (***), ns (not significant, $p > 0.05$) determined by Student's $t$ test (two experimental groups) or one-way ANOVA (more than two experimental groups). Cytokine ratio data were evaluated using two-tailed Mann–Whitney test.

plays a critical role in protecting against immune pathology in vivo via induction of the immune-modulatory cytokine IL-10.

Chronic LCMV infection represents the prototypic model to study T cell exhaustion, where simultaneous co-expression of several inhibitory receptors marks T cells with diminished effector function[34,43]. We found that antibody-mediated TIGIT blockade during chronic LCMV infection resulted in marked downregulation of other co-inhibitory receptors, such as PD-1, Tim-3, and Lag-3 on total CD8$^+$ T cells and on CD4$^+$ T cells. In contrast to CD8$^+$ T cells, where co-inhibitory receptor expression

was significantly lower throughout the duration of the chronic infection, PD-1 downregulation was only transient on CD4+ T cells and occurred during early stages of the infection. This might contribute to CD8+ dysfunction, since CD4+ help has been described to be essential for effective virus clearance in this system[44,45]. TIGIT blockade also modestly increased the production of TNF-α by CD8+ T cells. Consistent with our findings after TIGIT blockade, antibody-mediated TIGIT stimulation correlated with the concomitant induction of PD-1 on CD8+ T cells, which supports the notion that TIGIT at least partially contributes to the development of T cell exhaustion. Moreover, we found that early T cell activation was significantly reduced after agonistic TIGIT ligation in vivo, which correlated with the considerably reduced production of pro-inflammatory cytokines. This is in line with previous reports demonstrating the potential of TIGIT to attenuate T cell responses[9,15,31].

Moreover, we were able to establish that TIGIT ligation in vivo results in enhanced induction of the immunosuppressive cytokine IL-10 in response to acute LCMV challenge. Vice versa we could verify that TIGIT blockade leads to the diminished expression of IL-10 by CD8+ T cells during chronic infection, which indicates that TIGIT is involved in shifting the cytokine balance toward a more IL-10 dominated immune response in the context of viral infections. Reports from different groups including our own support this concept, as TIGIT expression was demonstrated to correlate with IL-10 expression in a model of antigen-specific tolerance[46], as well as peptide immunization[9,31]. Notably, TIGIT blockade alone was insufficient to promote viral clearance in the LCMV clone 13 model, which is consistent with previous reports showing that only co-blockade of TIGIT and PD-L1 restores the cytokine production of exhausted T cells and enhances viral clearance[8,47]. Surprisingly, TIGIT blockade resulted in the loss of antigen-specific T cells, which could account for the failure of the anti-TIGIT treatment to promote viral clearance. One potential mechanism that might contribute to lower cell counts could be enhanced cell death as a consequence of hyperactivation of these cells. Activation-induced cell death (AICD) occurs after repeated stimulation of the TCR and is important for maintaining immune homeostasis. AICD has also been described to be a major cause of T cell depletion in other chronic infections, such as HIV[48,49], independent of direct cytotoxic effects of the virus itself. Intriguingly, Tim-3 and Lag-3 also appear to be dispensable for the development of T cell exhaustion, as Tim-3 KO and Lag-3 KO mice were demonstrated to harbor fewer antigen-specific T cells and do not control virus replication after challenge with chronic LCMV clone 13[50,51]. In contrast, PD-1/PD-L1 blockade during chronic viral infections restores the function of exhausted CD8+ T cells, in particular antigen-specific cells, and results in enhanced T cell responses and virus control in different virus models[8,52,53], which suggests that these two molecules operate through distinct pathways to dampen immune cell responses and might thus partake in different physiological processes, where immune suppression is required.

Strikingly, lack of PD-1/PD-L1 signaling has been shown to be associated with pronounced immune pathology and increased mortality very early after systemic LCMV clone 13 infection[42,52,54]. Given that TIGIT−/− mice do not develop spontaneous signs of autoimmunity[8,9], it is tempting to speculate that TIGIT possesses a more localized role that is subordinate to PD-1. LCMV is known to inflict severe liver pathology as a consequence of virus-specific CD8+ T cell activation[39,55], and remarkably, we found that antibody-mediated TIGIT stimulation significantly reduced the secretion of liver-derived enzymes AST and ALT into the blood. What is more, we confirmed that TIGIT engagement protected acutely infected mice from weight loss and restricted leukocyte infiltration and inflammation in the liver. These findings could also be validated in an influenza model. Whether this also translates into improved survival in infections with lethal doses of virus will have to be addressed in future studies. Our study showed that engagement of the TIGIT pathway limited infection-induced liver and lung damage in an IL-10-dependent manner, thus revealing a critical mechanistic aspect of TIGIT biology. The fact that the ameliorated liver pathology was dependent on IL-10 is consistent with established knowledge showing that IL-10 in particular is essential for negative immune cell regulation[56]. The vital role of IL-10 in protecting against immune pathology becomes especially apparent during CNS infections, such as Japanese Encephalitis virus infection or coronavirus-induced encephalitis, where IL-10 producing T cells have been demonstrated to improve survival by ameliorating immunopathology[57,58]. In the context of influenza infection, however, the role of IL-10 still remains controversial as the genetic background appears to have a dominant effect on disease susceptibility[59,60]. More recently, Smith et al. demonstrated that IL-10 can act directly on CD8+ T cells by increasing the antigenic threshold for T cell activation and that this process was highly dependent on the dose and timing of exposure[17]. Since the impact of IL-10 is clearly determined by the timing and site of its production, TIGIT modulation might be used as a surrogate to generate a local immune suppressive environment at sites of tissue inflammation. Our improved understanding of the local effects of checkpoint inhibitors could thus feed into the development of novel therapies for the treatment of inflammatory diseases, where off target effects are encountered often and interfere with treatment success. TIGIT stimulation might thus be especially useful for prolonging graft survival or for managing side effects of checkpoint inhibitor therapy in combination with other checkpoint inhibitors. In conclusion, our results extend previous knowledge of the importance of TIGIT in suppressing immune cell activation and show a more specialized tissue-protective role of TIGIT in response to viral challenges.

## Methods

**Mice.** C57BL/6 (B6) mice were purchased from Janvier Labs. Congenic Ly5.1[61] and Thy1.1[62], Foxp3-GFP.KI[63] and Thy1.1-IL-10 reporter mice[30], P14[64], Smarta[65], IL-10[−/−][66], and PKOB mice have been described previously and were all on C57BL/6 background. Mice were used between 8 and 16 weeks of age and were sex and age matched within experiments. All animals were bred and housed in SPF and OHB facilities at LASC Zürich, Switzerland. All experiments were performed in accordance with institutional ethical policies and national regulations and have been reviewed and approved by the Cantonal veterinary office of Zurich.

**Viruses and infections.** The LCMV Clone 13 strain was propagated on BHK21 cells. Briefly, BHK21 cells were infected in suspension using a MOI of 0.1 (5 ml) for 1 h at RT in T175 tissue culture flasks (TPP) in serum-free modified Eagle medium (Gibco) on a plate shaker at minimum speed. After 1 h, 15 ml of fresh modified Eagle medium (Gibco) supplemented with 5% FCS (Corning) and 1% L-glutamine (Gibco) were added and incubated at 37 °C and 10% $CO_2$ for 48 or 72 h. Supernatants were frozen at −80 °C for long-term storage. Working stocks were generated by diluting frozen stocks in PBS. Animals were infected i.v. with either $2 \times 10^6$ FFU to induce chronic LCMV or with $1 \times 10^5$ FFU to induce an acute LCMV infection. For IAV infection, mice were infected with 200 PFU of PR8 (H1N1, $LD_{50} = 1250$ PFU) i.n. followed by antibody treatment on days 0, 2, 4 with either 100 μg of mouse IgG1 or anti-TIGIT antibody (1G9). On day 8 p.i., mice were injected i.v. with 200 μl of 0.5% Evans blue dye (EB, Sigma) in sterile 1× PBS 20 min before sacrifice. Animals were perfused with PBS, and the lung weight measured. Lungs were then placed in 1 ml Formamide (Sigma) and incubated at 56 °C overnight. The amount of extracted Evans blue (EB) dye was measured by photometry at 620 nm, and quantified against a standard curve using the following formula: EB concentration (ng/ml) × volume of formamide (ml)/total lung weight (g). Influenza virus was titrated by Plaque assay on MDCK cells.

**Antibodies and MHC class I tetramers.** The LCMV glycoprotein peptides gp33-41 (gp33 peptide, KAVYNFATM) and gp61-80 (gp61, GLNGPDIYKGVYQFKS-VEFD) were purchased from EMC microcollections GmbH. Allophycocyanin (APC)- conjugated peptide/MHC class I tetrameric complexes were generated as previously described[67]. The following anti-mouse monoclonal antibodies were used

from Biolegend: CD4 (RM4-5 or GK1.5), PD-1 (29 F.1A12), CD44 (IM7), CD8 (53-6.7), IFN-γ (XMG1.2), Lag-3 (C9B7W), TIGIT (1G9), TNF (MP6-XT22). CD45.1 (A20) and CD90.1 (OX-7), CD49b (DX5), CD69 (H1.2F3), NKG2D (CX5), CD11b (M1/70), MHCII (M5/114.15.2), Ly6C (HK1.4), Ly6G (1A8), NK1.1 (PK136), F4/80 (BM8). TCF-1/TCF7 (C63D9) was purchased from Cell Signaling Technology. Tim-3 (215008) was purchased from R&D Systems. Granzyme B (GB11) was purchased from Thermo Fisher. Dilution factors for the antibodies used in this study can be found in Supplementary Table 1. The anti-TIGIT mAbs for blockade (clone 1B4) and stimulation (clone 1G9) were described previously[31]. Control animals were treated with mouse IgG1 (BioXCell).

**Ex vivo stimulation and cytometric bead assay**. Single-cell suspensions were generated by mechanical disruption in RPMI 1640 medium (Gibco) supplemented with 10% FCS (Corning), penicillin (100IU/ml, Gibco) and 1% L-glutamine (Gibco). Red blood cells were removed by adding ACK lysis buffer (155 mM $NH_4Cl$, 10 mM $KHCO_3$, 0.1 mM $Na_2EDTA$, pH: 7.4) for 3 min. Splenocytes were stimulated for 48 h in the presence of anti-CD3 (1 μg/ml) without additional anti-TIGIT antibodies. Supernatants were stored at −20 °C for subsequent cytokine analysis by cytometric bead assay (CBA). BioLegend's LEGENDplex kit and the BD CBA kit were used to detect soluble IL-10 protein according to the manufacturers' instructions. The raw data were analyzed with LEGENDplex software (Biolegend) or FCAP Array software (Soft Flow Inc.) to calculate unknown protein levels against a known standard curve.

**In vivo antibody treatment**. Mice were injected i.p. with either 100 μg mouse IgG1 (BioXCell) or anti-TIGIT Abs (1G9 or 1B4 clone) as described previously[31]. Injections were given on days 0, 2, 4, 10, 17, and 24 after infection if not stated otherwise during chronic LCMV infection. The treatment regimen was shortened accordingly during acute infections.

**Adoptive cell transfers**. Total $CD4^+$ T cells or $CD8^+$ T cells were pre-sorted from spleens of naive Smarta mice and P14 TCR-α/β mice respectively using the MojoSort mouse CD4 or CD8 T cell isolation kit (Biolegend) according to the manufacturers' instructions. In total, $1 \times 10^4$ cells per cell type were adoptively transferred i.v. into C57BL/6 (B6) recipient mice. The following day, the recipient mice were infected i.v. with $2 \times 10^6$ FFU LCMV Clone 13.

**Focus-forming assay LCMV**. Virus titers were determined as described previously[68]. Briefly, MC57G cells were seeded at $5 \times 10^5$ cells/ml in 24 well plates in modified Eagle medium supplemented with 5% FCS and 1% PSG. Organ samples were homogenized by mechanical disruption using the TissueLyserII (Qiagen) and centrifuged for 1 min at 4 °C at full speed. Samples were serially diluted onto the MC57G cell layer and overlaid with 2% methylcellulose dissolved in 5% DMEM medium (Gibco). The assay was incubated for 48 h at 37 °C in 5% $CO_2$. After 2 days, cells were fixed with 4% PFA for 30 min, followed by 20 min permeabilization with 1% Triton X-100. The cell layer was washed twice with 1× PBS, followed by 1 h incubation with 10% FCS in PBS at RT. The primary antibody (VL-4 rat anti-LCMV mAb) was added 1:25 in PBS and incubated for 1 h at RT. Cells were then washed twice with 1× PBS and incubated with the secondary antibody peroxidase-conjugated goat anti-rat IgG (Jackson ImmunoResearch) 1:400 in 10% FCS in PBS. OPD (Sigma-Aldrich) was added according to the manufacturer's instructions and incubated for at least 30 min at RT.

**Plaque assay influenza A virus**. To determine the amount of infectious IAV particles, Madin–Darby Canine Kidney (MDCK) cells (ATCC CCL34) were seeded into 12-well plates in Dulbecco's Modified Eagle Medium (Life technologies) supplemented with 10% fetal bovine serum (Life technologies), 1% L-glutamine (Life technologies), and 1% penicillin/streptomycin (Life technologies) at 37 °C in 5% $CO_2$. At 100% cell confluency, cells were washed once with PBS supplemented with 20 μM $Mg^{2+}$, 10 μM $Ca^{2+}$, 0.3% (w/v) bovine serum albumin (BSA), and 1% (v/v) penicillin–streptomycin (infection PBS) before being infected with a series of tenfold dilutions of virus containing sample in infection PBS. After 1 h incubation at 37 °C in 5% $CO_2$, the inoculum was removed, and the cells were covered with an agar overlay containing 0.7% (v/v) Oxoid agar (Sigma-Aldrich) dissolved in Modified Eagle Medium (Life Technologies) supplemented with 10 μg/ml DEAE Dextran (Sigma-Aldrich), 0.1% (v/v) sodium bicarbonat (Sigma-Aldrich) and 1 μg/ml tosylamide-2-phenylethyl-chloromethyle-ketone (TPCK)-treated trypsin (Sigma-Aldrich). Cells were incubated at 37 °C in 5% $CO_2$ until plaques were visible. Then, cells were fixed with 3.7% PFA for 20 min at RT before staining with 0.5% (w/v) crystal violet (Sigma-Aldrich) in $ddH_2O$ with 20% methanol for 10 min.

**Flow cytometry**. FACS stainings were performed on single-cell suspensions from the spleen, liver, and lung. The spleen samples were prepared by mechanical disruption in RPMI 1640 medium supplemented with 10% FCS, penicillin (100IU/ml) and 1% L-glutamine. Liver and lung were enzymatically digested for 30 min, and immune cells isolated using a 30% Percoll (GE Healthcare) gradient. Red blood cells were removed by adding ACK lysis buffer (155 mM $NH_4Cl$, 10 mM $KHCO_3$,

0.1 mM $Na_2EDTA$, pH: 7.4) for 3 min. Cells were re-stimulated with either anti-CD3 (1 μg/ml), gp33 peptide or gp61 peptide (EMC microcollections) and Bre-feldin A solution (Biolegend) at 37 °C in 10% $CO_2$ for 3–4 h before staining. For surface stainings, antibodies were incubated for 20–30 min at RT in PBS. For intracellular cytokine staining, spenocytes were permeabilized using the Cytofix/Cytoperm kit (BD Biosciences) for 5 min at RT, followed by antibody incubation for 20–30 min at RT. For staining of transcription factors, cells were stained and permeabilized with the Foxp3/Transcription factor staining buffer set (eBioscience). The Zombie NIR fixable dye was used to exclude dead cells and debris. Data were acquired on a BD LSR Fortessa or BD FACS CantoII analyzer (BD Bioscience) and analyzed using Flowjo software (TreeStar). Gating strategies are shown in Supplementary Fig. 9.

**Histopathology**. Liver and lung lobes from IgG1 or anti-TIGIT Ab (1G9)-treated and uninfected mice were fixed in 4% paraformaldehyde and transferred to 70% EtOH for storage before processing. The organs were trimmed, and routinely paraffin wax embedded. Consecutive sections (3–5 μm) were prepared, and routinely stained with hematoxylin-eosin (HE) or subjected to the Periodic Acid Schiff (PAS) reaction for the demonstration of hepatocellular glycogen accumulation. Livers were assessed for any histopathological changes including leukocyte infiltration in a single blinded experimental setup, where the pathologist was unaware of the clinical status and treatment that the mice had received. Histopathological changes were graded on the basis of the assessment schemes published by Ishak et al.[69] and Thoolen et al.[70]. The following histological features were considered: leukocyte infiltration in portal areas, around central vein and as random focal aggregates, and hepatocyte death (necrosis, apoptosis); the distribution of changes was characterized as focal, multifocal, or diffuse (grading from 0 to 3).

**Analysis of serum liver AST and ALT**. Serum AST and ALT were measured at the Clinical Chemistry Department of the University Hospital, Zurich, Switzerland. The catalytic concentration of both enzymes was measured at 37 °C with pyridoxal phosphate activation on a Roche Cobas 8000 (c502) according to IFCC.

**Statistical analysis**. Statistical analyses were performed using Graphpad Prism. Appropriate statistical tests were selected as indicated in the figure legends with significant differences marked on all figures. Differences between control and treatment groups were determined using two-tailed Student's $t$ test, including Welch's correction for unequal variants if necessary. To compare more than two groups, we used one-way ANOVA with Tukey's multiple comparisons test. Histopathological data was evaluated using two-tailed Mann–Whitney test. Significance was defined as $p < 0.05$ (*), $p < 0.01$ (**), $p < 0.005$ (***), or ns (not significant, $p > 0.05$).

**Reporting summary**. Further information on research design is available in the Nature Research Reporting Summary linked to this article.

## Data availability
The data that support the findings of this study are available from the corresponding author upon reasonable request. Raw data for all figures are provided as a Source Data file.

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

## Acknowledgements

We would like to thank Casey Weaver for the Thy1.1-IL-10 reporter mouse strain, Vijay Kuchroo for anti-TIGIT antibodies, Nina Henle, Helen Thut, and Catharina Küng for technical assistance, the Joller and Oxenius group members for helpful discussions. We are also grateful to the laboratory technicians in the Histology Laboratory, Institute of Veterinary Pathology, Vetsuisse Faculty, University of Zurich, for excellent technical support. This work was supported by the Swiss National Science Foundation (PP00P3_150663 and PP00P3_181037 to N.J.), the European Research Council (677200 Immune Regulation to N.J.), the Zuercher Universitaetsverein (ZUNIV-FAN to N.J.), the Olga Mayenfisch Stiftung (to N.J.), and the Novartis Foundation for medical-biological research (17A027 to N.J.).

## Author contributions

Conceptualization: M.S. and N.J. Experimentation: M.S., N.R., K.L., A.H., and K.P. Analysis: M.S., A.H., and A.K. Writing—original draft: M.S. and N.J. Review and editing: all authors. Funding acquisition: N.J. Supervision: A.O., S.S., and N.J.

## Competing interests

The authors declare no competing interests.
