## [Peer Review File · Nature Communications]

Reviewers' comments:

Reviewer #1 (Remarks to the Author):

Schorer et al., reported that TIGIT expression is associated with higher IL-10 production in mice infected by LCMV. Modulation of TIGIT by blocking and agonistic antibodies changed the phenotype and cytokine production of T cells, but did not affect viral persistence. In particular, TIGIT expression is associated with higher IL-10 production, which efficiently suppresses immune mediated liver damage in acute LCMV infection. Moreover, similar effect was observed in H1N1 influenza A virus infection. Co-inhibitory receptors are critical to control excess immune activation. However, the functional difference of co-inhibitory receptors remains unclear. This study clearly showed that TIGIT is effective to suppress immune pathology through increased IL-10 expression. These findings are critical to understand exact roles of co-inhibitory receptors in complex immune system, and lead to novel therapy of immune mediated diseases, including autoimmune diseases, and chronic viral infection.

Major comments

1. In this study, agonistic anti-TIGIT antibody (1G9) was used to study TIGIT mediated signaling. However, its signaling differs from natural signaling thorough TIGIT-CD155 interaction since TIGIT stimulates CD155 expressing cells through reverse signaling. TIGIT promotes the generation of immunoregulatory dendritic cells (Yu X, et al., Nat Immunol, 10: 48-57, 2009). Authors should examine IL-10 production from non-T cells including dendritic cells during LCMV infection.
2. Elevated AST, and ALT levels were very low in IL-10 KO mice after LCMV infection. On the other hand, the results using influenza virus showed that IL-10 is involved in immune pathology using IL-10KO mice. Authors should discuss about the mechanism why loss of IL-10 suppresses liver damage caused by LCMV.
3. TIGIT can induce IL-10 production from various cells. Conditional knockout of IL-10 will reveal which cells are responsible for IL-10 mediated suppression of immune pathology.

Minor concerns

1. In line 136 of page 5, "fro" is "from".
2. In line 164 of page 6, "(1B4)" should move after blocking anti-TIGIT antibody".
3. In line 272, "Figure 6J" should be "Figure 6I".

Reviewer #2 (Remarks to the Author):

The manuscript by Schorer et al is a straight forward manuscript, which reveals the role of TIGIT during acute and chronic virus infection. Specifically, the manuscript determines how TIGIT influences IL-10 production and PD-1 expression in T cells and thereby modulate CD8 T cell function, virus control and immunopathology. The authors nicely dissect the role of TIGIT on PD-1 expression and IL-10 production. They made the surprising discovery that TIGIT specifically modulates virus induced immunopathology but not virus control. While this is a very interesting and important study, some more points have to be clarified to draw final conclusions and make the paper suitable for publication in nature communications.

Major points:

Many of the data are made with gain of function and lack of function experiments using blocking and activating antibodies. However the virus titers were only analyzed with lack of function while the immunopathology was analyzed with gain of function. Is there a rational behind this? While obviously these are taff experiments it might nevertheless make sense to analyze the phenotype of CD8 T cells using antagonistic antibody. This would allow to directly link a T cell phenotype to the immunopathology phenotype.

The authors nicely determine the role of TIGIT antibodies in virus infection. It seems obvious that

T cells are the major target cells, which are modulated by TIGIT. However there might be a role of NK cells. Is TIGIT expressed in NK cells? Were NK cell numbers and functions analyzed +/- TIGIT treatment?

The authors have not really explained why immunopathology but not virus control is modulated by TIGIT. Is it simply because immunopathology is more sensitive to TIGIT, or are specific functions, which are mainly responsible for immunopathology, in CD8 T cells modulated by TIGIT?

Minor points:

Figure 1: Original FACS blot would be nice to see; This would allow to judge whether TIGIT is shifting up during infection, or whether TIGIT+ cells are a distinct population.

Figure 3D: it remains unclear whether Blocking Ab (1B4) was added also in culture. If not, it might be that the secondary expansion is additionally affected by TIGIT blockade. Here it would have been nice to control for numbers and activation status of T cells within the cultures.

It is nice that the color code is the same throughout the manuscript. Still sometimes the Figures were hard to follow. I recommend a graphic summary after acceptance of the manuscript, to conclude the most relevant data.

Reviewer #3 (Remarks to the Author):

The current study by Shorer et al., investigates the role of TIGIT in both persistent and acute viral infection using LCMV and Influenza mouse models. The main theme of the study is employing anti-TIGIT agonistic and antagonistic antibodies during both acute and persistent infection to assess the resulting T cell responses and viral control/immune pathology. The authors demonstrate that blocking TIGIT can have immune pathological consequences during acute LCMV infection despite minimal effects during persistent infection both with respect to T cell rescue and viral control. However, the most interesting data comes in the final figure where the authors demonstrate that treatment with an anti-TIGIT agonistic antibody can rescue immune pathology during acute influenza virus infection, which appeared dependent on IL-10 to some degree although the data is quite weak and confusing at times.

Major comments:

1. The first half of the paper is incremental at best as others have published that TIGIT blocking antibodies fail rescue T cell control of persistent viral infection alone and require checkpoint blockade. One interesting piece of data from the first half of the manuscript was the fact that blocking TIGIT resulted in reduced virus-specific CD8 T cells however, the reason for this was not followed up in this manuscript.
2. Investigating how anti-TIGIT agonist antibody protected from immune pathology was more interesting however, there were some significant issues with the data produced.
 - The weight loss in Figure 6A was very modest compared to what multiple labs across the globe have reported with medium dose ($1-2 \times 10^5$ PFU Clone13 IV). This dose of infection routinely produces 20-30% weight loss and some death. Was death observed in these experiments? Why is the weight loss so minimal?
 - Based on the authors introduction, TIGIT has been linked to IL-10 production and thus a natural connection would be to link the pathology to less IL-10 however, IL-10KO mice displayed reduced liver damage as measured by ALT/AST serum levels which were similar to agonist TIGIT treatment. This is quite confusing and does not seem to support the hypothesis.
 - It would be important to measure viral loads in all the infection models as enhanced viral control

could account for the observed results.

3. The influenza data in the presence of TIGIT agonism was the most interesting of the results however, again many important parameters were left out.

- What is the lethal dose 50 for the PR8 strain used in this study? Can TIGIT agonism prevent death following lethal PR8 infection?

- What about viral titers? Does TIGIT agonism prevent virus control? A thorough viral titer kinetics should be done.

- The authors measure IL-10 producing T cells using the Thy1.1 IL-10 reporter mouse. What are the actual IL-10 levels in the BALF/Lung homogenate following TIGIT agonism?

- Does TIGIT blockade exacerbate immune pathology during influenza virus infection? What about viral clearance?

- Lung histology should also be determined.

- Figure 7A shows increased leakage of EB into lung in IL-10KO during TIGIT agonist antibody treatment however, there is only 2 mice in this group. The n should be increased here like observed in the other groups. Keeping with this point, why is there less leakage in IL-10 KO mice? Based on papers by Sun and Braciale (*Nature medicine*, 2009, PMID 19234462) there should be more leakage and lung damage. This paper should also be cited.

Point-by-point reply

Manuscript NCOMMS-19-17946A: “TIGIT limits immune pathology during viral infections”; Schorer et al.

We would like to thank the reviewers for their constructive criticism. We believe that we were able to address all of their concerns and think the additional data has further strengthened our finding that TIGIT limits immune pathology in both LCMV and influenza viral infection models and allowed us to establish a causal link between the observed effects on immune pathology and cytotoxicity. The detailed description on how we addressed all issues raised is outlined in the point-by-point response below.

Reviewer #1 (Remarks to the Author):

Major comments

1. In this study, agonistic anti-TIGIT antibody (1G9) was used to study TIGIT mediated signaling. However, its' signaling differs from natural signaling thorough TIGIT-CD155 interaction since TIGIT stimulates CD155 expressing cells through reverse signaling. TIGIT promotes the generation of immunoregulatory dendritic cells (Yu X, et al., *Nat Immunol*, 10: 48-57, 2009). Authors should examine IL-10 production from non-T cells including dendritic cells during LCMV infection.

We agree with the reviewer that this is an interesting point and have now examined IL-10 production in myeloid cell populations during both acute and chronic LCMV infection upon treatment with TIGIT modulating antibodies (Supplementary Figure 3 of the revised manuscript). We couldn't detect any significant differences in the myeloid compartment upon anti-TIGIT treatment during early phases of both infections except for slightly higher frequencies of IL-10-Thy1.1⁺ cells within the CD11b⁺Ly6C⁺ cell population after agonistic anti-TIGIT (1G9) Ab treatment. However, the overall frequency of IL-10-Thy1.1⁺ cells in this cell population is much smaller than in the T cell compartment and thus likely has a minimal impact on the overall response.

2. Elevated AST, and ALT levels were very low in IL-10 KO mice after LCMV infection. On the other hand, the results using influenza virus showed that IL-10 is involved in immune pathology using IL-10KO mice. Authors should discuss about the mechanism why loss of IL-10 suppresses liver damage caused by LCMV.

The maintenance of chronic LCMV infection critically depends on IL-10 as IL-10^{-/-} mice or mice in which IL-10R was blocked effectively clear Clone 13 LCMV infection (Brooks et al., *Nature Medicine*, 2006; Richter et al., *European Journal of Immunology*, 2013; Ejrnaes et al., *Journal of Experimental Medicine*, 2006). Lack of IL-10 therefore significantly alters the course and virus kinetics of LCMV infection, which in turn affects the degree of inflammation and hence tissue damage. As such IL-10^{-/-} mice are able to rapidly clear LCMV Clone 13 and thus display reduced tissue damage at later time points. We have revised the manuscript to highlight this point and included a viral kinetic in WT and IL-10^{-/-} mice quantifying the virus load in the new Supplementary Figure 6. In contrast to LCMV, IL-10 depletion has no effect on virus clearance *in vivo* in the influenza infection model (Sun et al., *Nature Medicine*, 2009). The different outcomes that we observed using these two infection models therefore very likely result from the differential sensitivity of these viruses to IL-10.

3. TIGIT can induce IL-10 production from various cells. Conditional knockout of IL-10 will reveal which cells are responsible for IL-10 mediated suppression of immune pathology.

We agree with the reviewer that the analysis of conditional IL-10 KO mice would be an elegant way to determine the relevant source of IL-10 for limiting tissue pathology. However, it would also require excessive crossing, as we do not have any of these lines available in our facility. As such, while we think this is a relevant and interesting point, given the extensive delay in publication the acquisition and crossing of the relevant mouse strains would cause, we feel that this is beyond the scope of the current study.

Minor concerns

1. In line 136 of page 5, “fro” is “from”.
2. In line 164 of page 6, “(1B4)” should move after blocking anti-TIGIT antibody”.
3. In line 272, “Figure 6J” should be “Figure 6I”.

We thank the reviewer for noticing these errors and have corrected the manuscript accordingly.

Reviewer #2 (Remarks to the Author)

Major points:

1. Many of the data are made with gain of function and lack of function experiments using blocking and activating antibodies. However, the virus titers were only analyzed with lack of function while the immunopathology was analyzed with gain of function. Is there a rational behind this? While obviously these are taff experiments it might nevertheless make sense to analyze the phenotype of CD8 T cells using antagonistic antibody. This would allow to directly link a T cell phenotype to the immunopathology phenotype.

We agree with the reviewer that addressing this point is important. We have now also included experiments analyzing immunopathology in the lack of function set-up, which is compiled in Supplementary Figure 8 of the revised manuscript. TIGIT blockade did not result in exacerbated immunopathology or disease in either of the models. Indeed, this is in line with our previous finding that lack of TIGIT does not lead to the development of spontaneous autoimmunity (Joller et al., *Journal of Immunology*, 2011). More strikingly though, TIGIT blockade results in the loss of antigen-specific cells (Figure 5), which might account for the lack of disease exacerbation.

A recent study found CD8⁺ T cell-mediated perforin secretion to be the cause of tissue damage in a model of viral hepatitis (Welz et al., *Nature Communications*, 2018). In keeping with this finding, we found that the tissue protective effect of TIGIT agonism was lost in PKOB mice, suggesting that the pathological process inhibited by TIGIT is perforin dependent (new Figure 6B-C of the revised manuscript).

2. The authors nicely determine the role of TIGIT antibodies in virus infection. It seems obvious that T cells are the major target cells, which are modulated by TIGIT. However, there might be a role of NK cells. Is TIGIT expressed in NK cells? Were NK cell numbers and functions analyzed +/- TIGIT treatment?

We thank the reviewer for this important comment. We have now analyzed the NK phenotype and numbers during both acute and chronic LCMV infection (new Supplementary Figure 2) but couldn't detect any striking differences in NK numbers or phenotype. Furthermore, based on the comment of reviewer 1 we have also examined the myeloid compartment to take into account that TIGIT is able to backsignal into APCs and anti-TIGIT antibody treatment might alter this process but also couldn't detect any relevant changes

(new Supplementary Figure 3). As such, our additional data confirms that T cells are the main cell population affected by anti-TIGIT treatment.

3. The authors have not really explained why immunopathology but not virus control is modulated by TIGIT. Is it simply because immunopathology is more sensitive to TIGIT, or are specific functions, which are mainly responsible for immunopathology, in CD8 T cells modulated by TIGIT?

It was previously shown that the CD8⁺ T cell driven immunopathology that arises in PD-L1 deficient mice after systemic LCMV infection is a consequence of perforin-mediated cytolysis (Frebel et al., *Journal of Experimental Medicine*, 2012). A more recent study found CD8⁺ T cell-mediated perforin secretion to be the cause of tissue damage in a model of viral hepatitis (Welz et al., *Nature Communications*, 2018). Therefore, we hypothesized that perforin might also play a critical role in the pathogenesis of immunopathology during acute LCMV infection. To test this, we infected PKOB KO mice with acute LCMV and treated them with IgG1 or agonistic anti-TIGIT antibody (1G9) and measured serum AST and ALT. We couldn't detect any differences between the treatment groups in PKOB KO mice, suggesting that the pathological process inhibited by TIGIT is perforin dependent (new Figure 6B-C of the revised manuscript). We also tested for granzyme secretion in IgG or agonistic anti-TIGIT antibody (1G9) treated CD8⁺ T cells by FACS and found that granzyme B production is significantly decreased already very early (d2) after acute LCMV infection (view Reply Figure 1).

Reply Figure 1: Granzyme B production in CD8⁺ T cells after antibody-mediated TIGIT stimulation. WT mice were infected with 1×10^5 FFU of LCMV clone 13 and either treated with IgG1 or agonistic anti-TIGIT antibody (1G9) on day 0 p.i. Representative FACS plots and a quantification of intracellular granzyme B production in CD8⁺ T cells is shown ($n=4$). Symbols in bar graphs represent individual mice. Statistical analysis using Mann-Whitney test.

Minor points:

Figure 1: Original FACS blot would be nice to see; This would allow to judge whether TIGIT is shifting up during infection, or whether TIGIT+ cells are a distinct population.

We agree with the reviewer and have added representative FACS plots of TIGIT expression on CD4⁺ T cells as Figure 1H.

Figure 3D: it remains unclear whether blocking Ab (1B4) was added also in culture. If not, it might be that the secondary expansion is additionally affected by TIGIT blockade. Here it would have been nice to control for numbers and activation status of T cells within the cultures.

No extra antibody was added to the cultures. We have rephrased this in the Materials and Methods section of the revised manuscript to make this clearer to the reader.

We agree with the reviewer that the numbers and activation status of cells should be considered. We analyzed the CD8⁺ T cell frequency, as well as their activation status (CD44

expression) on the day of organ harvest and couldn't detect any differences regarding these two parameters (see Reply Figure 2). What is more, overall T cell numbers in chronically LCMV infected mice are not altered as shown in Supplementary Figure 1. As no differences were observed in number, frequency, or activation status, no additional normalization was performed for these experiments.

Reply Figure 2: CD8⁺ T cell frequency and activation phenotype after chronic LCMV infection. WT mice were infected with 2×10^6 FFU of LCMV clone 13 and either treated with IgG1 or blocking anti-TIGIT antibody (1B4) on days 0, 2, 4, 10, 17, and 24 p.i. The frequency and activation status of CD8⁺ T cells isolated from the spleen on the days indicated is shown ($n=3-9$). Symbols in bar graphs represent individual mice. Data from 2-3 independent experiments is shown. Statistical analysis using Mann-Whitney test.

It is nice that the color code is the same throughout the manuscript. Still sometimes the Figures were hard to follow. I recommend a graphic summary after acceptance of the manuscript, to conclude the most relevant data.

We thank the reviewer for this comment and have added a graphic abstract to the submission documents.

Reviewer #3 (Remarks to the Author):

Major comments:

1. The first half of the paper is incremental at best as others have published that TIGIT blocking antibodies fail rescue T cell control of persistent viral infection alone and require checkpoint blockade. One interesting piece of data from the first half of the manuscript was the fact that blocking TIGIT resulted in reduced virus-specific CD8 T cells however, the reason for this was not followed up in this manuscript.

The loss of antigen-specific T cells after TIGIT blockade was indeed an interesting finding. It was reported previously that virus-specific TCF-1⁺CD8⁺ T cells are capable to sustain the immune response to chronic LCMV infection, because they give rise to differentiated cells that can respond to inhibitory receptor blockade (Utzschneider et al., *Immunity*, 2016). We found that TIGIT blockade leads to the loss of TCF-1⁺ CD8⁺ T cells *in vivo* during chronic infection in both endogenous and virus-specific CD8⁺ T cell populations, which might explain why TIGIT blockade has no beneficial effect on the anti-viral immune response. This additional data is found in Supplementary Figure 5 of the revised manuscript.

2. Investigating how anti-TIGIT agonist antibody protected from immune pathology was more interesting however, there were some significant issues with the data produced.
- The weight loss in Figure 6A was very modest compared to what multiple labs across the globe have reported with medium dose ($1-2 \times 10^5$ PFU Clone 13 IV). This dose of infection routinely produces 20-30% weight loss and some death. Was death observed in these experiments? Why is the weight loss so minimal?

As pointed out by the reviewer, some studies using 2×10^5 FFU LCMV clone 13 (Waggoner et al., *Journal of Virology*, 2014; Cornberg et al., *Frontiers in Immunology*, 2013) found weight loss ranging between 20-25%. However, other labs have reported more moderate weight loss in the range that we observed (Baazim et al., *Nature Immunology*, 2019: 5-10%). Furthermore, we used a lower dose of 1×10^5 FFU, in the range of another study, which also reported very moderate weight loss (5-8% weight loss, inoculation dose of 8×10^4 FFU of LCMV clone 13; Stamm et al., *Virology*, 2012). As such, our observations are very much in line with previous reports and matching the mild weight loss, we also did not observe any death in our experimental setup.

- Based on the authors introduction, TIGIT has been linked to IL-10 production and thus a natural connection would be to link the pathology to less IL-10 however, IL-10KO mice displayed reduced liver damage as measured by ALT/AST serum levels which were similar to agonist TIGIT treatment. This is quite confusing and does not seem to support the hypothesis.

We thank the reviewer for this comment pointing out that the reduced liver damage in IL-10 KO mice might be confusing. The persistence of chronic LCMV infection critically depends on IL-10 as IL-10^{-/-} mice or mice in which IL-10R was blocked effectively clear Clone 13 LCMV infection (Brooks et al., *Nature Medicine*, 2006; Richter et al., *European Journal of Immunology*, 2013; Ejrnaes et al., *Journal of Experimental Medicine*, 2006). Lack of IL-10 therefore significantly alters the course and virus kinetics of LCMV infection, which in turn affects the degree of inflammation and hence tissue damage. As such IL-10^{-/-} mice are able to rapidly clear LCMV Clone 13 and thus display reduced tissue damage at later time points. We have revised the manuscript to highlight this point and included a viral kinetic in WT and IL-10^{-/-} mice quantifying the virus load by plaque assay in the new Supplementary Figure 6.

- It would be important to measure viral loads in all the infection models as enhanced viral control could account for the observed results.

We agree with the reviewer that this is an important piece of information and have extensively expanded our data in this regard. In the revised manuscript we now include a detailed PR8 virus kinetic (compiled in Supplementary Figure 7A and 7B) and viral kinetics for acute LCMV infection (Supplementary Figure 6C). The detailed virus kinetic following blocking anti-TIGIT antibody (1B4) treatment in chronically LCMV infected mice is compiled in Figure 5G and 5H. TIGIT agonism does not prevent virus control during PR8 infection (Supplementary Figure 7A and 7B) and it only minimally and transiently affects virus loads very early during acute LCMV infection (d5) in the spleen, but not in the liver (Supplementary Figure 2E).

3. The influenza data in the presence of TIGIT agonism was the most interesting of the results however, again many important parameters were left out.

- What is the lethal dose 50 for the PR8 strain used in this study? Can TIGIT agonism prevent death following lethal PR8 infection?

We received the viral preparation that we are using from a collaborator who had determined the LD₅₀ to be 1250 PFU. Unfortunately, our animal license only allows for infection with 50-

200 PFU of influenza PR8 and hence we are not able to determine the effect of TIGIT agonism in a (sub-)lethal infection setting. However, we agree with the reviewer that this is an interesting question and have included it in the discussion of the revised manuscript.

- What about viral titers? Does TIGIT agonism prevent virus control? A thorough viral titer kinetics should be done (int dose CI13 LCMV and PR8).

As pointed out in response to one of the preceding questions (last section of question 2), we agree with the reviewer that this is an important piece of information and have amended data on the viral kinetics for PR8 (Supplementary Figure 7) and LCMV infection (Supplementary Figure 6) in the revised manuscript.

- The authors measure IL-10 producing T cells using the Thy1.1 IL-10 reporter mouse. What are the actual IL-10 levels in the BALF/Lung homogenate following TIGIT agonism?

We thank the reviewer for this important comment and have complemented our data from IL-10 reporter mice, which allow us to determine the cellular source of IL-10, by measuring IL-10 protein levels in lung homogenates of IAV PR8 infected mice treated either with control IgG or agonistic anti-TIGIT (1G9) antibody (Figure 7G of the revised manuscript). Matching the flow data obtained with the reporter mice, we confirmed higher IL-10 levels in the lungs of mice treated with the agonistic anti-TIGIT antibody.

- Does TIGIT blockade exacerbate immune pathology during influenza virus infection? What about viral clearance?

Based on this comment as well as on a comment by reviewer 2 (major point 1), we complemented our manuscript with an analysis of the effect of TIGIT blockade in both infectious conditions (LCMV and IAV PR8) in order to elucidate if TIGIT blockade exacerbates immune pathology. As shown in Supplementary Figure 8, TIGIT blockade results in delayed recovery from chronic LCMV infection-induced body weight loss (Supplementary Figure 8A). However, this does not go along with increased enzyme leakage from the liver (Supplementary Figure 8B). Similarly, we couldn't detect any differences in AST and ALT secretion early during acute LCMV infection (Figure 8C). Moreover, we analyzed pro- and anti-proinflammatory cytokine levels in lung homogenates after IAV PR8 infection and didn't find any striking differences between control IgG1 and blocking anti-TIGIT antibody (1B4) treated organs (Supplementary Figure 8D). Along the same lines, histological examination of lung slides did not reveal exacerbated disease in this model (Supplementary Figure 8E and Supplementary Figure 7F). We also quantified virus loads in the lungs of IAV PR8 infected animals with and without TIGIT blockade and found no significant differences (Supplementary Figure 8F). In light of the fact that TIGIT KO mice do not develop spontaneous autoimmunity (Joller et al., *Journal of Immunology*, 2011), these findings are not surprising. Furthermore, since TIGIT blockade leads to the loss of antigen-specific cells in the LCMV model (Figure 5 A-C), it seems straightforward to assume that the immune pathology is not exacerbated because of this loss of highly activated cells.

- Lung histology should also be determined.

We agree with the reviewer and have included a lung histopathological analysis in Supplementary Figure 7C-F of the revised manuscript. The histological analysis confirmed our findings using Evan's blue in that mice treated with agonistic anti-TIGIT antibody showed reduced vasculitis and leukocyte recruitment.

- Figure 7A shows increased leakage of EB into lung in IL-10KO during TIGIT agonist antibody treatment however, there is only 2 mice in this group. The n should be increased here like observed in the other groups. Keeping with this point, why is there less leakage in IL-10 KO mice? Based on

papers by Sun and Braciale (Nature medicine, 2009, PMID 19234462) there should be more leakage and lung damage. This paper should also be cited.

We thank the reviewer for catching this, we have updated Figure 7A to include additional data points (n=6). As the reviewer correctly noticed, the level of lung leakage we observe is comparable between WT and IL-10KO mice. Sun et al. (*Nature Medicine*, 2009) have indeed shown that blockade of IL-10R signaling during influenza challenge leads to lethal pulmonary inflammation. However, their experimental conditions differed from what we report here in that we use a different mouse strain (C57BL/6 vs. Balb/c) and a relatively low dose of 150 FFU influenza PR8, which only results in minor weight loss and limited pathology, while Sun et al. infected with substantially higher doses of virus (500 PFU). Furthermore, it is important to note that conflicting reports exist on the impact of IL-10 on immune pathology during influenza infection as several other groups have demonstrated that IL-10 deficiency enhances survival after influenza challenge (McKinstry et al., *Journal of Immunology*, 2009; Sun et al., *Journal of Virology*, 2010). Importantly however, the survival curves of WT and IL-10KO in all reports only start to diverge after one week of infection (see Sun et al. 2009, Figure 5a; Sun et al. 2010, Figure 2d; McKinstry et al. 2009, Figure 1a, b), which is the endpoint of our study. We have modified the discussion to include this point and included the relevant references.

By incorporating the experiments suggested by the reviewers into the revised manuscript we were able to further strengthen the finding that TIGIT limits immune pathology in an IL-10-dependent manner without affecting viral clearance. We provide additional data connecting TIGIT modulation with cytotoxicity and reveal that TIGIT blockade results in a loss of TCF-1⁺ CD8⁺ T cells, which are essential for recovery of CTL function of exhausted cells upon checkpoint blockade. Furthermore, extending our analysis of the effects of TIGIT modulation to innate cells allowed us to confirm that T cells are the main cell population affected by anti-TIGIT treatment. We hope that with these additional the data, our manuscript is now suitable for publication in *Nature Communications*.

Reviewers' comments:

Reviewer #1 (Remarks to the Author):

The authors responded to the comments except conditional KO experiment. This question should be addressed in the future.

Reviewer #2 (Remarks to the Author):

All concerns raised were addressed extensively and strengthen the main conclusions of the manuscript. I recommend publication without further delay.

Reviewer #3 (Remarks to the Author):

While the reviewers have addressed some of the minor concerns of my previous critique, the data still remains relatively weak as far as confirming that TIGIT agonism has a significant effect on immune pathology during influenza virus infection. The most interesting aspect of the manuscript was the influenza virus studies however, the authors really don't conclusively show that TIGIT has a significant effect on the pathological response following influenza infection.

Point-by-point reply

Manuscript NCOMMS-19-17946A: “TIGIT limits immune pathology during viral infections”; Schorer et al.

We would like to thank the reviewers for their positive feedback and are pleased that reviewers 1 and 2 consider our manuscript suitable for publication. The last open point comes from reviewer 3:

Reviewer #3 (Remarks to the Author):

While the reviewers have addressed some of the minor concerns of my previous critique, the data still remains relatively weak as far as confirming that TIGIT agonism has a significant effect on immune pathology during influenza virus infection. The most interesting aspect of the manuscript was the influenza virus studies however, the authors really don't conclusively show that TIGIT has a significant effect on the pathological response following influenza infection.

Our paper conclusively shows that TIGIT engagement limits immune pathology in viral infections without affecting viral load. We demonstrate that this tissue protective effect is IL-10-dependent and have confirmed our results in two independent viral infection models, namely LCMV and influenza infection, highlighting the general nature of our finding. Furthermore, with the revisions we have performed in response to the reviewer's initial comments, we have extensively expanded our data set in influenza infection. Specifically, we have:

- performed viral kinetic and conclusively show that TIGIT agonism doesn't affect viral titers in influenza infection (Supplementary Figure 7A, B).
- complemented our cytokine analysis by FACS using IL-10 reporter mice with measurements of IL-10 protein in the lung (Figure 7G).
- expanded our cytokine analysis by FACS to include TNF- α in addition to IFN- γ and IL-10 (Figure 7F, Supplementary Figure 8D).
- performed lung histology (Supplementary Figure 7C-F) and confirmed the differences in pathology observed by Evan's Blue staining (Figure 7A, B).
- increased the group size for the assessment of lung permeability by Evan's Blue and fully confirm our finding that TIGIT engagement reduces tissue damage upon influenza infection in an IL-10-dependent manner (Figure 7A).

The only point we were not able to address was the question whether TIGIT agonism can prevent death following lethal influenza infection. As pointed out in our previous reply, we agree with the reviewer that this is an interesting question. However, as also pointed out, our animal license does not allow for performing infections with lethal doses, but is limited to infections with 50-200 PFU influenza PR8 (all animal licenses in Switzerland are only granted in this specific manner). To perform lethal infections we would have to obtain a new license, which would take approx. 6 months given that it would involve infections with a lethal dose, which usually take several rounds of revisions with the authorities to obtain. Given the minor knowledge gain that this would allow for, we do not think that this justifies such a significant delay.

While we agree that it would be an interesting experiment to perform, there is no basis for the assumption that the outcome would be different than at lower infectious doses. Increasing the infectious dose might enhance the T cell response but we already observe the same tissue protective effect upon infection with LCMV, which is one of the most potent inducers of T cell responses. Thus, there is no reason to assume that increasing the infectious dose of influenza would alter the outcome of TIGIT engagement.

In conclusion, we have comprehensively demonstrated that TIGIT engagement limits immune pathology in influenza and LCMV infection without affecting viral clearance and that this process is dependent on IL-10.